# Concepts in Motion: Temporal Bottlenecks for Interpretable Video Classification

## Abstract

Conceptual models such as Concept Bottleneck Models (CBMs) have driven substantial progress in improving interpretability for image classification by leveraging human-interpretable concepts. However, extending these models from static images to sequences of images, such as video data, introduces a significant challenge due to the temporal dependencies inherent in videos, which are essential for capturing actions and events. In this work, we introduce **MoTIF** (Moving Temporal Interpretable Framework), an architectural design inspired by a transformer that adapts the concept bottleneck framework for video classification and handles sequences of arbitrary length. Within the video domain, concepts refer to semantic entities such as objects, attributes, or higher-level components (e.g., "bow," "mount," "shoot") that reoccur across time—forming motifs collectively describing and explaining actions. Our design explicitly enables three complementary perspectives: global concept importance across the entire video, local concept relevance within specific windows, and temporal dependencies of a concept over time. Our results demonstrate that the concept-based modeling paradigm can be effectively transferred to video data, enabling a better understanding of concept contributions in temporal contexts while maintaining a competitive performance.

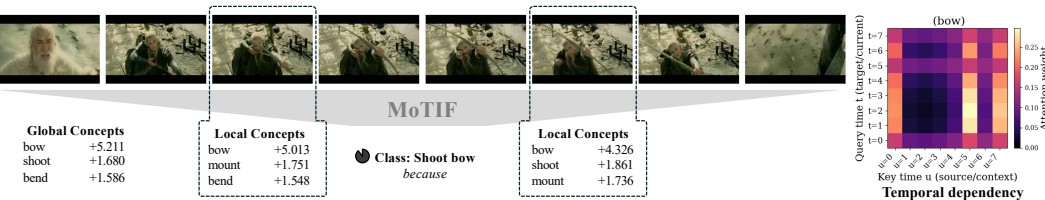

Figure 1: **MoTIF.** The framework takes videos as input and produces local concept explanations for local windows, global explanations for entire videos, and temporal dependency maps from the attention heads of the transformer module. Model represents MoTIF (ViT-L14) and sample frames are from HMDB51 (Kuehne et al., 2011), licensed under CC BY 4.0.

## 1 Introduction

Modern deep learning models already achieve outstanding results in video understanding tasks such as video classification, action recognition, and event detection (Liu et al., 2021; Bertasius et al., 2021). Despite their success, these models are commonly perceived as *black boxes* since their internal workings are not interpretable in a way that reveals their decision-making process (Molnar et al., 2020; Knab et al., 2025b). Concept Bottleneck Models (CBMs) (Koh et al., 2020) address this issue by enforcing an intermediate bottleneck layer of human-understandable concepts, which are then used by a linear classifier to generate the final prediction.

While CBMs have been extensively studied in the image domain (Prasse et al., 2025; Yang et al., 2023; Sun et al., 2025; Schrodi et al., 2025), their extension to video remains largely unexplored (Jeyakumar et al., 2022). Videos differ from images in that they contain a *temporal component*: concepts evolve over time and many actions cannot be deduced from a single frame (Lee et al., 2025b; Chen et al., 2025). In addition, the length of a video is not fixed, making the adaptation of

existing CBM architectures non trivial, since they always require the same embedding dimensionality. Transformer-based models (Vaswani et al., 2017) are powerful for modeling such long-range dependencies (Bertasius et al., 2021), but their dense feature mixing hinders clear attribution at the concept level, making their interpretation infeasible (Hao et al., 2021; Molnar et al., 2020).

In this work, we introduce **MoTIF** (**Mo**ving **T**emporal **I**nterpretable **F**ramework), a concept bottleneck model tailored for video classification. MoTIF builds on transformer-inspired blocks and introduces a *per-channel temporal self-attention* (diagonal attention) mechanism that isolates temporal reasoning for each concept via depthwise $1\times1$ convolutions. To illustrate how MoTIF extends beyond static images, Fig. 1 shows our framework: it processes a video, tracks its concepts through time, and explains which concepts drive the final prediction. In this example, the model identifies reoccurring semantic motifs such as a *bow* being *mounted* and the arrow being *shot* by Legolas (Lord of the Rings), which together form the basis for recognizing the higher-level action "shooting a bow." Temporal dependency maps further reveal that specific concepts primarily attend to characteristic frames, i.e., the moments when the bow is mounted and the arrow released. As this example shows, the design preserves concept separation across time and enables analysis of both *global concept contributions* over an entire video and *local relevance* within specific frames.

Our key contributions are:

- A CBM framework, *MoTIF*, for video sequences that **supports arbitrary-length inputs** and integrates seamlessly with vision–language backbones.

- Per-channel temporal self-attention that **preserves concept independence** within transformer blocks and models temporal dynamics on a per-concept basis.

- *MoTIF* is the first method to **enable three complementary explanation modes**: *(i)* global concept relevance via log-sum-exp (LSE) pooling, *(ii)* localized temporal explanations using windowed concept attributions, and *(iii)* attention-based temporal maps that visualize how a concept channel distributes its focus across time.

Extensive experiments and ablations (window size, backbone selection, temperature $\tau$, full vs. diagonal attention, etc.) demonstrate that MoTIF achieves robust performance while providing fine-grained and faithful explanations.

## 2 METHOD

In this section, we present the MoTIF framework for interpretable video classification. The goal is to preserve concept-level transparency while leveraging temporal modeling capabilities.

**MoTIF** is a CBM with a transformer architecture for video classification that operates directly on sequences of per-window concept activations (Figure 2). A video is represented by $T$ temporal windows, each described by $C$ scalar concept activations, with $k = |C|$; the input is $X \in \mathbb{R}^{T \times C}$.

The framework enforces strict *per-concept interpretability*: temporal dependencies are modeled independently for each concept channel, so that recurrent temporal patterns—*motifs*—remain attributable to individual concepts. After temporal processing, per-concept activations are refined via a nonnegative affine transformation, and a classifier produces per-time-step logits. Video-level predictions are obtained via log-sum-exp pooling, which also yields a time-importance profile for additional explanation. In contrast, a conventional transformer mixes channels during attention (Vaswani et al., 2017). For comparison, we also evaluate a variant with *full multi-head attention*, but emphasize that this removes explicit concept attribution. A detailed analysis of this and other design choices is provided in Section 3.2.2 and Appendix C. For completeness, an algorithmic overview of the framework and its subparts is included in Appendix D.

**Channels as concepts.** We use the term *channel* to denote one dimension of the concept space. Each video $n$ yields $y^{(n)} \in \mathbb{R}$ classes, $X^{(n)} \in \mathbb{R}^{T_n \times C}$ concept activations, where the second axis ($C$) indexes concepts; each slice $X^{(n)}_{:,c} \in \mathbb{R}^{T_n}$ corresponds to the temporal activation sequence of concept $c \in C$. Unlike CNN channels, these dimensions are semantically interpretable by design.

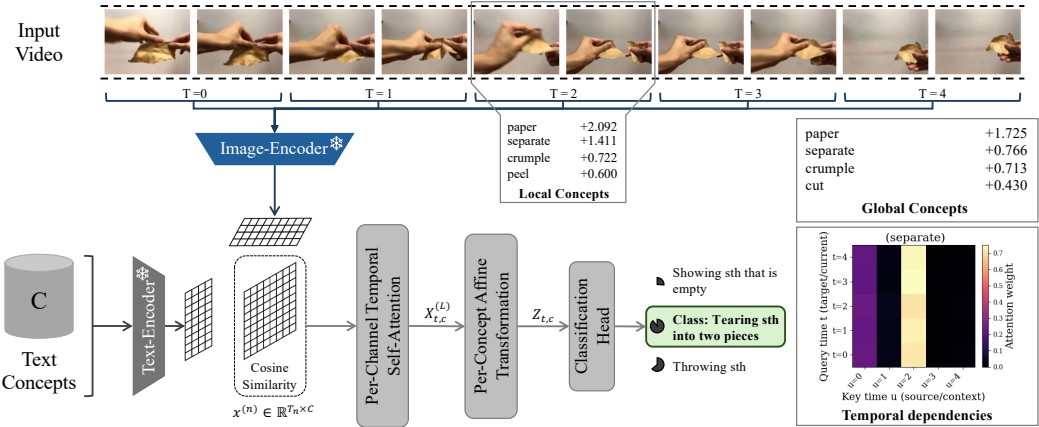

Figure 2: **MoTIF pipeline.** Videos are embedded with a vision–language backbone and mapped to concept activations via cosine similarity. Per-channel temporal self-attention models dynamics independently for each concept, followed by a nonnegative affine transformation and classification. MoTIF enables explanations across three views: global concepts, local concepts, and temporal dependencies. Sample frames from SSv2 (Materzynska et al., 2020) with MoTIF (ViT-L14).

**Notation.** Capital and lowercase letters denote matrices and vectors, respectively. During training, a batch of videos is represented as

$$X \in \mathbb{R}^{B \times T \times C}, \tag{1}$$

where $B$ is the batch size, $T$ is the (padded) sequence length, and $C$ is the number of concepts. Thus $X_{:,:,c} \in \mathbb{R}^{B \times T}$ contains the activations of concept $c$ across the batch.

## 2.1 TRANSFORMER BOTTLENECK MODEL

**Video and concept embeddings.** We employ an image–text aligned backbone (Radford et al., 2021) to map frames into a shared embedding space. Window embeddings are obtained by a random frame in the window or via a video-adapted CLIP variant (see Appendix A.3), and concept activations are computed as cosine similarities to a defined concept bank, producing $\{X^{(n)}\}_{n=1}^{N}$ with $X^{(n)} \in \mathbb{R}^{T_n \times C}$. The concept bank $C$ is constructed from natural language descriptions of human-interpretable actions and objects. To generate a diverse and semantically rich vocabulary, we use a large language model (LLM) (OpenAI et al., 2024) (GPT-5) to propose candidate concepts, following Yang et al. (2023) who leverage LLMs for automated concept set generation. Since concept generation is not the main focus of this work, we adopt this approach without further refinement.

### 2.1.1 PER-CHANNEL TEMPORAL SELF-ATTENTION

In standard transformers, query–key–value (QKV) projections are implemented as full linear layers ($W_Q \in \mathbb{R}^{C \times C}$), which mix channels and would obscure concept attribution. In MoTIF, we avoid mixing: each concept $c$ receives its own QKV projections via depthwise $1 \times 1$ convolutions,

$$Q, K, V \in \mathbb{R}^{T \times C}, Q_{:,c} = x_{:,c}\,\theta_Q^{(c)},\ K_{:,c} = x_{:,c}\,\theta_K^{(c)},\ V_{:,c} = x_{:,c}\,\theta_V^{(c)}, \tag{2}$$

with $\theta_{Q,K,V}^{(c)} \in \mathbb{R}$. Here $\theta_Q^{(c)}, \theta_K^{(c)}, \theta_V^{(c)}$ are channel-specific scalars applied uniformly across all $T$, not temporal filters. Equivalently, each projection uses a depthwise kernel $\Theta_Q, \Theta_K, \Theta_V \in \mathbb{R}^{C \times 1 \times 1}$, where $\theta_*^{(c)}$ is the $c$-th depthwise filter. Thus, temporal information is preserved and channels remain independent, without cross-channel mixing. Attention scores are computed *per concept* as $W_{c,t,u} = Q_{t,c} K_{u,c}$, where $t$ denotes the query time step and $u$ the key/value time step. A softmax over $u$ yields attention weights $W \in \mathbb{R}^{C \times T \times T}$, so that each concept decides *which of its past or future activations to attend to*. The output is obtained as the weighted sum over $V$:

$$X_{t,c}^{(L)} = \sum_{u=1}^{T} W_{c,t,u}\,V_{u,c}. \tag{3}$$

The difference between full and diagonal attention is visualized in Equation 4 as the structure of the concept–concept attention matrix. That is the reason why we call this mechanism *diagonal attention*: each channel learns to construct its own temporal filter by weighting past activations differently at each step. Unlike fixed convolutional kernels, the attention weights adapt to the input sequence, yet the restriction to depthwise (per-concept) projections guarantees that evidence for one concept does not leak into another.

$$
\underbrace{\begin{bmatrix}
(c_1 \to c_1) & (c_1 \to c_2) & \cdots & (c_1 \to c_C) \\
(c_2 \to c_1) & (c_2 \to c_2) & \cdots & (c_2 \to c_C) \\
\vdots & \vdots & \ddots & \vdots \\
(c_C \to c_1) & (c_C \to c_2) & \cdots & (c_C \to c_C)
\end{bmatrix}}_{\text{Full attention}}
\quad
\underbrace{\begin{bmatrix}
(c_1 \to c_1) & & & \\
& (c_2 \to c_2) & & \\
& & \ddots & \\
& & & (c_C \to c_C)
\end{bmatrix}}_{\text{Diagonal attention}}
\tag{4}
$$

The block concludes with per-channel normalization and a lightweight feed-forward network (two depthwise $1 \times 1$ convolutions with GELU and dropout). For MoTIF, stacking more blocks does not add richer hierarchical abstractions, because cross-concept interactions (which normally grow expressivity in deep transformers) are deliberately suppressed.

**Architectural extension.** For most experiments in this paper, we report the results of MoTIF using diagonal attention within a standard transformer architecture. However, as shown by Bertasius et al. (2021), separating spatial and temporal attention can further enhance performance. Therefore, we additionally evaluate MoTIF when extended to a space-time transformer architecture, as detailed in Appendix A.4. This variant achieves even higher performance, demonstrating that MoTIF is not restricted to a single transformer design.

**Complexity.** Diagonal attention reduces the channel-mixing cost from $\mathcal{O}(C^2 T)$ to $\mathcal{O}(CT)$, but it requires computing a full $T \times T$ attention map for every channel, yielding $\mathcal{O}(CT^2)$. By contrast, standard multi-head attention only maintains a fixed number of maps (one per head), scaling as $\mathcal{O}(HT^2)$ with $H \ll C$. Thus, with many concepts, full attention is often more efficient, while diagonal attention deliberately trades computational cost for strict concept isolation.

### 2.1.2 PER-CONCEPT AFFINE TRANSFORMATION

Each refined activation $X_{t,c}^{(L)}$ can optionally be scaled and shifted by learnable concept-specific parameters, $\tilde{X}_{t,c} = \gamma_c X_{t,c}^{(L)} + \delta_c$, and then passed through a Softplus nonlinearity $Z_{t,c} = \text{Softplus}(\tilde{X}_{t,c})$, which ensures nonnegative concept activations while avoiding dead units and maintaining differentiability everywhere. This transformation introduces a per-concept scale ($\gamma_c$) and bias ($\delta_c$), allowing the model to adapt to differences in concept magnitude and activation thresholds.

### 2.1.3 CLASSIFICATION HEAD

From these activations, per-time-step logits are computed as $\ell_t = W_k Z_{t,:} + b$ with $W_k \in \mathbb{R}^{K \times C}$, where $K$ denotes the number of target classes and $C$ the number of concepts. Since videos vary in length, we apply *log-sum-exp (LSE) pooling* across time (Wang et al., 2018), which smoothly interpolates between mean-pooling($\tau \to 0$) and max-pooling ($\tau \to \infty$):

$$
\hat{c} = \tfrac{1}{\tau} \log \sum_{t=1}^{T} m_t e^{\tau c_t}, \qquad \hat{\ell} = \tfrac{1}{\tau} \log \sum_{t=1}^{T} m_t e^{\tau \ell_t},
\tag{5}
$$

where $m_t \in {0, 1}$ are masks for padded windows. We denote the pooled concept vector by $\hat{c}$ and the pooled logits by $\hat{\ell}$. The pooled logits $\hat{\ell}$ form the video-level prediction.

**Training objective.** The model is trained with class-weighted cross-entropy on $\hat{\ell}$, complemented with two regularizers: an $\ell_1$ penalty on $W$ to encourage sparsity, and an activation sparsity penalty on $Z$:

$$
\mathcal{L} = \text{CE}(\hat{\ell}, y) + \lambda_{\ell_1} \|W_k\|_1 + \lambda_{\text{sparse}} \frac{1}{(\sum_t m_t) C} \sum_{t,c} m_t |Z_{t,c}|.
\tag{6}
$$

## 2.2 EXPLANATION

To make predictions transparent, MoTIF decomposes them into time- and concept-resolved contributions. For a target class $k$, the contribution of each time step is $c_t^{(k)} = Z_{t,:} W_{k,:}$, with score $s_t^{(k)} = \sum_{c=1}^{C} c_{t,c}^{(k)} + b_k$. Temporal importance weights are derived consistently with LSE pooling:

$$\pi_t^{(k)} = \frac{\exp(s_t^{(k)}/\tau)}{\sum_{u=1}^{T} \exp(s_u^{(k)}/\tau)}. \tag{7}$$

Aggregating over time yields global concept attributions, $\bar{c}^{(k)} = \sum_{t=1}^{T} \pi_t^{(k)} c_t^{(k)}$. MoTIF therefore provides three complementary views: **(1) Global concepts**, via $\bar{c}^{(k)}$; **(2) Local concepts**, active in high-weight windows (large $\pi_t^{(k)}$); **(3) Temporal dependencies**, revealed by per-concept attention maps ($W_{c,t,u}$) that show how occurrences of concepts relate across time (see Appendix B.3). Together, these expose both *which* concepts mattered and *when* they were decisive.

## 3 EXPERIMENTS

In this section, we evaluate MoTIF on known video classification benchmarks. We then analyze different architectural design choices through ablation studies, illustrate the kinds of explanations our model provides with representative examples, and finally examine its behavior under interventions.

### 3.1 EXPERIMENTAL SETUP

**Datasets.** We evaluate MoTIF on four established benchmarks: Breakfast Actions (Kuehne et al., 2014), HMDB51 (Kuehne et al., 2011), UCF101 (Soomro et al., 2012), and Something-Something V2 (SSv2) (Goyal et al., 2017; Materzynska et al., 2020). These datasets contain 10, 51, 101, and 174 classes, respectively. These datasets span diverse settings: short versus long clips, local versus global temporal dependencies, and varying degrees of abstraction and viewpoint diversity.

**Backbones.** We adopt a range of CLIP-based backbones differing in model size, patch resolution, and temporal adaptation as feature extractors. Specifically, we include RN/50, ViT-B/16, ViT-B/32, and ViT-L/14 from CLIP (Radford et al., 2021), SigLIP (Zhai et al., 2023), and video-adapted Perception Encoder (PE) (Bolya et al., 2025). SigLIP replaces CLIP's contrastive softmax with a pairwise sigmoid loss, decoupling image and text encoders, which improves training efficiency and scalability. The Perception Encoder is trained on video–text pairs and incorporates a mechanism to aggregate frame-level information. For both PE and SigLIP we use the ViT-L/14 backbone.

**Concepts.** For concept creation, we follow Yang et al. (2023) and prompt an LLM (GPT-5) (OpenAI et al., 2024) to generate unique textual concepts per dataset; these concepts are further described in Appendix E.

**Baselines.** In line with (Rao et al., 2024; Prasse et al., 2025), we evaluate MoTIF against zero-shot and linear-probe baselines. To ensure a fair comparison with MoTIF, we compute zero-shot predictions on the window-level and apply majority voting across windows, preventing the zero-shot baseline from relying solely on a single global aggregation. As a supervised baseline (Global CBM), we train a linear classifier on top of mean-pooled window embeddings, analogous to a global bottleneck. This setup captures only coarse video-level information, in contrast to MoTIF, which models temporally localized concept activations. All experiments were conducted on a single NVIDIA A6000 GPU with 48 GB of VRAM.

### 3.2 EXPERIMENTAL RESULTS

#### 3.2.1 PERFORMANCE EVALUATION

We report Top-1 accuracies of MoTIF with different backbones across four datasets in Table 1, using the uniform hyperparameters described in A.1. For each dataset, we provide the mean and standard deviation over the available test splits (see Appendix C.1). The global CBM consistently outperforms zero-shot, while MoTIF surpasses both. Accuracy increases with backbone capacity, indicat-

Table 1: **Performance comparison (% Top-1 accuracy).** Mean $\pm$ standard deviation on train-test splits on Breakfast Actions, HMDB51, UCF101, and SSv2 with different CLIP-based backbones. We report seconds per training epoch next to the accuracy scores for all MoTIF variants.

| Method | Breakfast | HMDB51 | UCF101 | SSv2 |
|---|---|---|---|---|
| *Zero-shot* | | | | |
| CLIP-RN/50 (Radford et al., 2021) | $18.6 \pm 2.6$ | $29.8 \pm 0.5$ | $57.2 \pm 0.9$ | 0.8 |
| CLIP-ViT-B/32 (Radford et al., 2021) | $23.2 \pm 2.9$ | $38.1 \pm 0.3$ | $59.9 \pm 0.4$ | 0.9 |
| CLIP-ViT-L/14 (Radford et al., 2021) | $31.1 \pm 4.7$ | $45.7 \pm 0.1$ | $70.6 \pm 0.5$ | 0.7 |
| SigLIP-L/14 (Zhai et al., 2023) | $23.6 \pm 5.0$ | $49.3 \pm 0.8$ | $80.4 \pm 1.4$ | 1.3 |
| PE-L/14 (Bolya et al., 2025) | $41.4 \pm 7.0$ | $56.7 \pm 0.6$ | $74.6 \pm 0.9$ | 2.2 |
| *Global CBM* | | | | |
| CLIP-RN/50 (Radford et al., 2021) | $36.5 \pm 9.0$ | $59.3 \pm 0.8$ | $80.0 \pm 0.7$ | 13.7 |
| CLIP-ViT-B/32 (Radford et al., 2021) | $37.2 \pm 9.1$ | $61.6 \pm 1.6$ | $82.8 \pm 0.7$ | 15.2 |
| CLIP-ViT-L/14 (Radford et al., 2021) | $55.3 \pm 10.2$ | $68.4 \pm 0.5$ | $90.0 \pm 1.1$ | 18.1 |
| SigLIP-L/14 (Zhai et al., 2023) | $57.1 \pm 10.9$ | $65.0 \pm 2.1$ | $90.5 \pm 0.5$ | 19.6 |
| PE-L/14 (Bolya et al., 2025) | $72.9 \pm 10.3$ | $74.4 \pm 0.6$ | $94.5 \pm 0.6$ | 25.5 |
| *MoTIF (ours)* | | | | |
| MoTIF (RN/50) | $52.8 \pm 6.9$ (4.2) | $62.8 \pm 1.1$ (0.9) | $82.8 \pm 0.6$ (1.5) | 16.0 (10.0) |
| MoTIF (ViT-B/32) | $53.4 \pm 6.9$ (4.2) | $65.3 \pm 1.8$ (0.9) | $85.6 \pm 1.2$ (1.5) | 17.5 (9.9) |
| MoTIF (ViT-L/14) | $69.3 \pm 6.2$ (4.3) | $73.3 \pm 1.0$ (0.8) | $93.2 \pm 0.7$ (1.5) | 20.4 (10.1) |
| MoTIF-ST (ViT-L/14) | $70.7 \pm 7.7$ (7.2) | $74.8 \pm 1.0$ (1.8) | $93.8 \pm 0.9$ (3.3) | 26.0 (26.8) |
| MoTIF (SigLIP-L/14) | $73.5 \pm 8.6$ (4.2) | $73.2 \pm 2.4$ (0.8) | $94.0 \pm 0.8$ (1.5) | 22.4 (9.8) |
| MoTIF (PE-L/14) | $\underline{83.6} \pm 6.5$ (4.3) | $\underline{79.6} \pm 0.3$ (0.9) | $95.4 \pm 0.7$ (1.5) | 30.0 (10.4) |
| MoTIF-ST (PE-L/14) | $\mathbf{84.1} \pm 6.4$ (7.3) | $\underline{79.6} \pm 0.7$ (1.8) | $96.3 \pm 0.6$ (3.3) | 35.1 (26.7) |
| *Non-interpretable video models* | | | | |
| TSM (Lin et al., 2019) | $59.1^1$ | 73.5 | 95.9 | 61.7 |
| No frame left behind (Liu et al., 2021) | $62.0^1$ | $73.4^1$ | $\underline{96.4}^1$ | $\underline{62.7}^1$ |
| VideoMAE V2 (Wang et al., 2023) | – | **88.1** | **99.6** | **76.8** |

ing that introducing a concept bottleneck in video models preserves—and can improve—predictive performance while adding interpretability.

Regarding the transformer architecture, both variants perform well, but the space-time version (MoTIF-ST) (Bertasius et al., 2021) consistently achieves significantly better results on SSv2. Moreover, variants such as SigLIP and PE achieve higher accuracy than their counterparts with comparable parameter counts. Performance is strong across datasets; SSv2 remains the most challenging due to its abstract classes (e.g., putting something onto something), which require MoTIF to capture similar relational representations. On Breakfast, MoTIF outperforms the global CBM by 6–16 percentage points; however, all methods (zero-shot, global CBM, and MoTIF) exhibit large standard deviations across test splits. For HMDB51, UCF101, and SSv2, the gap is smaller (0.9–8.2 percentage points), likely because these datasets consist of shorter clips, where a simple mean representation already suffices to identify the class, especially for the PE backbone, which shows the strongest performance for video embeddings. However, on some datasets there is still a performance gap compared to strong non-interpretable baselines such as TSM (Lin et al., 2019), NoFrameLeftBehind (Liu et al., 2021) and VideoMAE V2 (Wang et al., 2023), especially for SSv2. Nevertheless, on Breakfast, we exceeded the performance of two of the baselines that report scores. Importantly, our objective is not to surpass state-of-the-art benchmarks, but to demonstrate a novel MoTIF framework for video data that provides unique interpretability insights.

### 3.2.2 Ablations

We perform a series of ablation studies to evaluate the effect of key design choices in MoTIF. Unless stated otherwise, experiments on Breakfast Actions and HMDB51 use CLIP ViT-B/32 and RN/50 backbones (hyperparameters in Appendix A.1). These two were chosen to cover distinct inductive biases: ViT-B/32 as a transformer-based (Dosovitskiy et al., 2020) extractor and RN/50 as a ResNet CNN (He et al., 2016), allowing evaluation of MoTIF across architecture families. These two datasets differ in scale, variability, and action granularity, making them complementary testbeds for analyzing architectural contributions. Further ablations can be found in Appendix C.

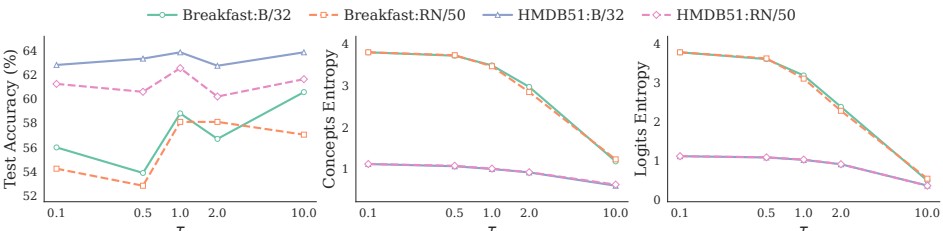

Figure 3: **Effect of log-sum-exp temperature $\tau$ on accuracy and entropy.** Accuracy is stable across $\tau$, while both concept- and logit-level entropy decrease as $\tau$ increases, yielding sharper time-importance distributions.

**Temperature $\tau$.** We vary the log-sum-exp pooling temperature $\tau$ to assess its effect on both accuracy and the sharpness of the time-importance distribution. Sharpness is quantified by the entropy of the softmax weights, computed either at the concept or at the logits level. Experiments show (see Figure 3) that accuracy varies only slightly across all tested values of $\tau$, indicating robustness of the predictive performance. In contrast, entropy decreases monotonically with larger $\tau$, i.e., the temporal weighting becomes more concentrated on fewer decisive frames. Hence, $\tau$ provides a controllable parameter: small values result in diffuse, high-entropy explanations, whereas large values produce sharp, low-entropy attributions, while accuracy remains relatively stable. Thus this parameter can be tuned by the user.

**Concept set influence.** To investigate the influence of the concept proposal set on the performance of MoTIF, we prompt GPT-5 five times, each time requesting a distinct set of candidate concepts (with some natural overlap across runs). All resulting concept sets for the five datasets are listed in Appendix E. Our experiments demonstrate that given the same prompt to the LLM, concept set affects test accuracy only moderately, with consistent trends across datasets and backbones (Fig. 4). In line with prior work on concept bottleneck models, more dataset-specific concepts tend to increase accuracy (Rao et al., 2024; Prasse et al., 2025; Schrodi et al., 2025). However, our main motivation here is not to optimize the quality of the concept proposals, but rather to demonstrate that our framework works broadly and robustly across datasets, backbones, and varying concept sets. In addition, we note that $k$—the number of con-

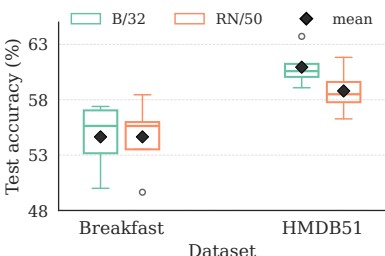

Figure 4: **Concept set influence.** Distribution of test accuracy across five different concept sets. The two dots indicate outliers within the interquartile range.

cepts retained in the bottleneck—influences the achievable accuracy: very small $k$ restricts expressive power, while very large $k$ may introduce noise and redundancy if concepts are too similar or irrelevant.

**Attention variant (full vs. diagonal).** While MoTIF enforces concept isolation, we also evaluate full multi-head attention; Figure 5 shows train and test accuracy for both. This test is designed to

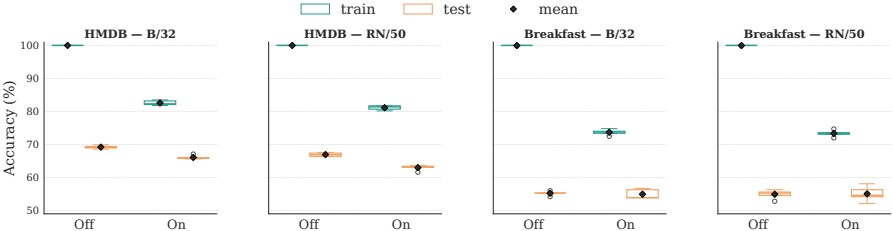

Figure 5: **Full vs. diagonal attention.** Train and test accuracy with and without enforcing diagonal attention over five seeds.

investigate if there are performance differences between those two versions. Comparable test accuracies with and without diagonal attention show that MoTIF performs similarly—though slightly lower due to the interpretability constraint—while only the training accuracy differs noticeably. However, additional experiments on SSv2 indicate that on more demanding datasets the effect becomes more pronounced, with test performance differences of up to 10.1%. We further illustrate the effect on explanations in B.2. When channels are mixed, the most important dimensions appear essentially random, severely reducing interpretability.

### 3.2.3 EXPLANATIONS

A central goal of our approach is to provide interpretable insights into the decision-making process of video classification models. We present further examples for the three explanation modes introduced in Section 2.2: global concept importance, local concepts in decisive windows, and temporal dependencies between concepts. In Figure 6, the first example from Breakfast Action shows preparing a *sandwich*. Global and temporal concepts highlight *bread* and *bagel*. The attention map for *bagel* reveals strong vertical stripes, meaning many query frames attend to the same key frames—anchor moments where the concept is most pronounced. Thus, *bagel* is localized in time but influences the entire sequence. The second example, from UCF101, depicts *kayaking*. Here, the most salient concepts are *paddle* and *kayak*, which remain stable over the short clip. The attention map for *paddle* shows a more uniform distribution across time, with slight emphasis at $u = 3$ and $u = 6$, indicating consistent expression of the concept rather than isolated peaks. Additional video examples are provided in the supplementary material (Appendix B.1 discusses misclassifications, and Appendix B.2 without diagonal attention), and all data will be released upon acceptance.

### 3.2.4 CONCEPT INTERVENTIONS

To illustrate the effect of concept manipulation, we revisit Figure 1. Ablating the most influential concept *bow* by zeroing its activations changes the prediction from the correct class to *run* (logit 8.20→6.79). Removing windows 1–4, where the bow is handled, shifts the output to *talk* (logit 6.75). In addition, we evaluate three interventions: (i) top-$k$ concept removal, (ii) random concept removal, and (iii) random insertion of noise into concept activations ($\epsilon \sim \mathcal{N}(0, \sigma^2)$ with $\sigma = 0.5$). On the dataset level, Table 2 shows that accuracy declines sharply only when the most influential concepts are removed, while random removal or noise insertion cause minimal degradation. This indicates that MoTIF's predictions rely on a small, semantically meaningful subset of concepts and remain robust under noise. Note that $k = 0$ corresponds to perfect accuracy, since all values are normalized relative to MoTIF's baseline predictions.

Table 2: **Concept interventions.** Accuracy after removing or perturbing $k$ concept channels (RN/50).

| Dataset | $k$ | Top-$k$ Removal | Random Removal | Random Insertion |
|---|---|---|---|---|
| Breakfast | 0 | 1.000 | 1.000 | 1.000 |
| | 1 | 0.496 | 0.989 | 0.979 |
| | 2 | 0.229 | 0.972 | 0.947 |
| | 3 | 0.085 | 0.951 | 0.951 |
| | 4 | 0.028 | 0.923 | 0.908 |
| HMDB51 | 0 | 1.000 | 1.000 | 1.000 |
| | 1 | 0.603 | 0.974 | 0.978 |
| | 2 | 0.374 | 0.963 | 0.961 |
| | 3 | 0.238 | 0.949 | 0.945 |
| | 4 | 0.142 | 0.935 | 0.921 |

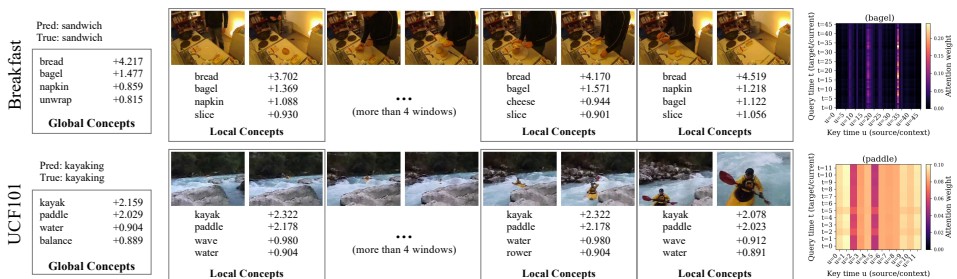

Figure 6: **MoTIF explanations.** Example videos from Breakfast and UCF101 with correct classifications, illustrating the three explanation modes supported by MoTIF (ViT-L14).

## 4  RELATED WORK

MoTIF positions itself at the intersection of three research areas: concept bottleneck models, temporal modeling, and video classification with vision–language backbones. At its core, MoTIF belongs to the family of concept bottleneck models (CBMs), which enforce intermediate, interpretable representations. At the same time, it incorporates transformer-based sequence modeling, situating it within the broader line of work on video classification and activity recognition. To the best of our knowledge, no prior work has combined these fields into a unified framework with three complementary views. Below, we review related work and outline how MoTIF extends these directions.

**Concept bottleneck models.**   CBMs (Koh et al., 2020) predict concepts as intermediate features before the final class prediction (Havasi et al., 2022; Chauhan et al., 2023; Prasse et al., 2025; Yang et al., 2023; Sawada & Nakamura, 2022). Recent research has focused on automatic concept discovery: DCLIP aligns data with CLIP's vision–language space (Menon & Vondrick, 2023), LaBo queries large language models for diverse candidate concepts (Yang et al., 2023), and DCBM leverages segmentation foundation models to extract object- and part-level concepts (Prasse et al., 2025). Other works have extended this idea to different modalities. For instance, Ismail et al. (2025) adapt the bottleneck principle to protein design, where interpretable biochemical features form the concept space. Wu et al. (2022) extend CBMs to time series data in the medical domain, demonstrating that clinically meaningful temporal features can act as concepts. More recently, Sun et al. (2025) apply the CBM framework to language models, where intermediate concepts correspond to interpretable linguistic or semantic units.

While CBMs have been widely explored in the image domain, their extension to sequential data has seen little attention. Jeyakumar et al. (2022) propose extracting concepts from video descriptions and using them as inputs to a CBM for video classification, but their approach operates on a global level and thus produces concept activations that are not tied to specific parts of a video. Lee et al. (2025a) propose DANCE, which disentangles video concepts by combining pose sequences with textual concepts, providing users with a two-sided comparison between textual concepts and motion-dynamic concepts. In contrast, MoTIF generalizes the bottleneck principle from static images to temporal sequences, enabling reasoning over evolving concept activations on established video benchmarks.

**Video classification and action recognition.**   Video classification not only assigns a label to an entire sequence but, in the case of action recognition, also requires identifying the actions occurring within it (Pareek & Thakkar, 2021). Both tasks have progressed from CNN-based architectures (Tran et al., 2015; Lin et al., 2019; Liu et al., 2021) to transformer-based (Wang et al., 2024a) models such as TimeSformer (Bertasius et al., 2021) and VideoMAE (Tong et al., 2022), which capture long-range dependencies and leverage large-scale self-supervised pre-training. MoTIF integrates these approaches by adapting a transformer block for temporal modeling while incorporating a 1D-CNN to enable per-channel computations.

**Temporal concept modeling.**   While concept-based explanations have been widely studied in the image domain (Knab et al., 2025a; Ghorbani et al., 2019), only a few works explore temporal dynamics (Gulshad et al., 2023; Kowal et al., 2024). Ji et al. (2023) propose a spatio-temporal concept framework to analyze representations in 3D ConvNets. PCBEAR (Lee et al., 2025b) introduces static and dynamic pose concepts for action recognition, and Saha et al. (2024) extend TCAV to videos by computing concept importance scores across sequences. 3D-ACE explains video models post hoc using user-defined concepts or supervoxels (Ghorbani et al., 2019). In contrast, MoTIF integrates concept-specific temporal attention directly into the predictive model, enabling reasoning over evolving concepts rather than treating them as fixed inputs or external explanations.

## 5  DISCUSSION

**MoTIF demonstrates that the CBM principle can be extended effectively to video data.** Our framework outperforms zero-shot classification and a global CBM that averages over windows, proving that the architecture yields not only interpretability but also improved predictive performance. MoTIF reaches competitive accuracy levels despite its explicit interpretability constraints.

Notably, MoTIF outperforms non-interpretable baselines (works that report results) on the Breakfast dataset, which contains the longest video sequences, and provides fine-grained insights by identifying the ordered steps in tasks such as meal preparation.

**MoTIF inherits the flexibility of CBMs.** In this work, we focus on CLIP-based variants and demonstrate that CLIP backbones trained on images remain effective when applied to video classification. We further show that adapted CLIP backbones, such as SigLIP or PE, achieve superior performance compared to their counterparts while maintaining the same parameter complexity. Especially PE that has been adapted to embed videos. Future studies could explore backbones tailored for video embeddings (Wang et al., 2024b), which must not necessarily be text-image aligned, e.g., follow the approach of (Prasse et al., 2025) or (Lee et al., 2025b). Moreover, the per-channel temporal self-attention block opens the possibility of applying MoTIF beyond video to other modalities with temporal structure, such as time series or text, or even other applications where channel preserving is important.

**Interpretability remains central.** MoTIF's three-level explanation property preserves concept representations without the entanglement issues that arise in standard attention mechanisms. Therefore, MoTIF gives valuable insights into the compositional structure of concepts the model has learned for its classification. However, the multiplication of concept activations with attention weights can sometimes yield unexpected outcomes, which complicates a direct comparison with standard action recognition benchmarks. Future adaptations that disentangle this effect may improve comparability.

**Several open challenges remain.** Selecting window size is non-trivial since actions span variable durations and can be missed if windows are too short, too long, or misaligned with action speed. Similarly, extracting richer and more action-relevant concepts directly from videos could improve performance, particularly for abstract datasets such as SSv2. In MoTIF, however, our objective was not to optimize for the most suitable concept set, but rather to demonstrate the functionality of the proposed architecture. One could also investigate how to model dependencies between concepts while maintaining interpretability. Capturing interactions is key for complex temporal reasoning, but care must be taken not to collapse into uninterpretable feature mixtures.

## 6 CONCLUSION

MoTIF, to the best of our knowledge, is the first concept bottleneck framework tailored to video data that identifies when and which concept is activated with diagonal attention. Through extensive evaluation across multiple embedding backbones and diverse datasets, we demonstrated its effectiveness and generality. Beyond strong empirical performance, MoTIF opens a new research direction by enabling fine-grained explanations of *which* visual *motifs* drive predictions in dynamic video settings. We believe this work can stimulate further exploration of concept-based models for video understanding, bridging the gap between interpretability and high-capacity temporal architectures.

### ETHICS STATEMENT

We follow the ICLR Code of Ethics. MoTIF aims to enhance the interpretability of video classification by exposing learned concepts in temporal sequences, thereby supporting more trustworthy AI. All datasets used are under licenses that permit scientific use and reporting of results.

### REPRODUCIBILITY STATEMENT

We provide code and examples with the submission. In addition, we provide an anonymous folder: https://gofile.io/d/T0QEzb, which contains an L/14 checkpoint of the trained MoTIF model with embedded data to replicate the results (HMDB51). MoTIF, its preprocessing and post-processing components are organized into separate files to facilitate navigation. Code follows the Python style guide for clarity. Notebooks are explicitly named to distinguish video handling, data embedding, model training, and explanation generation. Additional implementation details are documented in the accompanying README.

## USE OF LARGE LANGUAGE MODELS

We employed large language models as auxiliary tools during the preparation of this work. All scientific content, ideas, and analyses were created by the authors. LLMs (Copilot, GPT-5) were used for secondary tasks, including generating plots for visualization, improving code quality to better follow Python style guidelines and enhance reproducibility, and refining the textual presentation of the manuscript. In addition, we used GPT-5's deep search functionality to cross-check the literature on concept bottleneck models for video classification. This search did not reveal any relevant works beyond those identified in our manual review, but we report its use here for transparency.

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

## A    IMPLEMENTATION DETAILS

### A.1    HYPERPARAMETER SETTINGS

Unless noted otherwise, all experiments in the main paper follow these default hyperparameters:

- **Training.** 100 epochs, batch size 32, AdamW with learning rate $10^{-3}$ and weight decay $10^{-2}$.
- **Pooling.** Log-sum-exp pooling with fixed temperature $\tau = 1.0$. Although LSE permits tuning of sharpness, we kept $\tau$ constant for unbiased comparisons.
- **Regularization.** Both the $\ell_1$ penalty on classifier weights and the activation sparsity penalty are set to $10^{-3}$.
- **Architecture.** One Transformer layer with per-channel (diagonal) attention; classifier weights constrained to be nonnegative. As stated in Section 2, the diagonal attention faces the trade-off: interpretability through concept isolation versus representational power. Thus, depth (stacking of more than one per-channel temporal block) mainly repeats the same constrained operation, yielding limited or no performance gains.
- **Data.** Window size of 16 frames per temporal unit. For HMDB51 and Breakfast we report results on split *s1*, for UCF101 on *testlist01*, and for HMDB51 on *split1*. Class weighting is applied to mitigate imbalance.

Dataset-specific deviations: for SSv2 and UCF101 we use learning rate $10^{-4}$, $\ell_1$ penalty $10^{-4}$, and sparsity penalty $10^{-4}$; for SSv2, the non-negativity constraint in AdamW is disabled. HMDB51 uses a shorter window size of 8. For Breakfast, the window size is increased to 32 to account for longer videos.

All experiments were run with a random seed of 42 for reproducibility. Section C further analyzes the sensitivity to random seed choice.

These values serve as the baseline configuration; modifications for ablation studies are detailed in Section 3.2.2.

### A.2    COMPUTATION TIME AND COMPLEXITY

Table 3 reports GPU memory (MB), epoch time (s) and throughput (samples/s) for HMDB51 and Breakfast using the two ablated clip backbones. Results are shown for diagonal attention enabled (On) and disabled (Off).

Table 3: **Complexity overview.** GPU memory, epoch time and throughput for RN/50 and B32 backbones with diagonal attention On/Off.

| Dataset | Backbone | Setting | GPU memory (MB) | Epoch time (s) | Samples / s |
|---------|----------|---------|-----------------|----------------|-------------|
| HMDB51 | RN/50 | On | 1723 | 0.91 | 4084 |
| | | Off | 517 | 0.97 | 4063 |
| | B32 | On | 2289 | 0.90 | 4108 |
| | | Off | 1064 | 0.95 | 4071 |
| Breakfast | RN/50 | On | 10634 | 4.26 | 454 |
| | | Off | 736 | 0.51 | 3461 |
| | B32 | On | 11463 | 4.28 | 445 |
| | | Off | 1565 | 0.53 | 3445 |
| Average $T$ (train set) | | | HMDB51: 12.5 | | Breakfast: 65.4 |

For short videos (small $T$) epoch times remain nearly unchanged while GPU memory differs substantially. For longer sequences, as in Breakfast, disabling diagonal attention reduces memory use considerably; the runtime and memory gap grows with sequence length since complexity increases with $T$ (see Sec. 2) if $H << C$. The runtime across backbones is similar in magnitude; for example the best-performing perception encoder in MoTIF required 4.55 s. Video embedding time is excluded because embeddings are reusable across models and vary with the embedding backbone.

## A.3 BACKBONE INTEGRATION

**Image-based.** Image–text models such as CLIP (Radford et al., 2021) and SigLIP (Zhai et al., 2023) embed single images rather than videos. To adapt them, we divide each video into windows of $F$ frames and randomly select one frame per window. This frame is embedded and serves as the window representation. Random sampling ensures variation in which part of the window is captured.

**Video-based.** The Perception Encoder (Bolya et al., 2025) is explicitly tuned to aggregate information across frames, producing a pooled embedding for each window. Unlike the image-based backbones, we therefore use multiple frames per window. In our experiments, we consistently used eight frames per window, except for HMDB51where the window size itself was eight, so only four frames were sampled.

## A.4 SPACE-TIME ATTENTION EXTENSION

While the standard MoTIF architecture applies temporal attention independently for each concept, we extend it with a factorized space-time attention mechanism (Bertasius et al., 2021) that enables concept interactions while preserving interpretability.

The *CBMTransformerST* architecture factorizes attention into two sequential components: spatial attention across concepts at each time step, followed by the original per-channel temporal attention.

**PerTimeSpatialBlock.** This module computes attention across concepts at each time step independently. Given input features $X \in \mathbb{R}^{B \times T \times C}$, spatial attention produces attention scores $W_s \in \mathbb{R}^{B \times T \times C \times C}$ where $W_s[b, t, i, j]$ represents the attention weight from concept $i$ to concept $j$ at time step $t$.

Unlike temporal attention, which maintains concept isolation, spatial attention allows concepts to interact within each temporal frame. To preserve concept interpretability while enabling controlled spatial interaction, we employ three identity-preserving mechanisms:

- **Identity bias**: We add a bias term $\alpha_I = 1.0$ to the diagonal of the attention score matrix, encouraging self-attention and reducing cross-concept mixing.

- **Spatial gating**: The spatial attention output is scaled by a gating factor $\beta_s \in [0, 1]$ before being added to the residual connection: $X' = X + \beta_s \cdot \text{SpatialAttn}(X)$. With $\beta_s = 0.1$ (default), the spatial mixing contributes 10% to the output, preserving 90% of the original concept identity.

- **Per-channel FFN**: The feed-forward network in the spatial block uses per-channel convolutions (grouped by $C$), ensuring no cross-concept mixing occurs in the FFN, matching the design of the temporal block.

These mechanisms ensure that concepts remain separable and interpretable: the spatial mixing is controlled and identity-preserving, the subsequent per-channel temporal attention maintains diagonal structure, and the final concept activations $Z_{t,c}$ and spatial attention maps $W_s$ enable attribution to individual concepts while revealing their interactions.

**SpaceTimeBlock.** The block applies spatial attention first to enable concept interactions at each time step, followed by per-channel temporal attention that maintains diagonal structure.

This factorization reduces computational complexity from $\mathcal{O}(T^2C^2)$ for full space-time attention to $\mathcal{O}(TC^2 + CT^2)$, while producing separate interpretable attention maps for spatial ($W_s \in \mathbb{R}^{B \times T \times C \times C}$) and temporal ($W_t \in \mathbb{R}^{B \times C \times T \times T}$) components.

**Exemplary Explanations.** MoTIF with CBMTransformerST (MoTIF-ST) applies two sequential attention mechanisms—spatial followed by temporal—yielding two complementary attention matrices for interpretation.

- The spatial attention block captures concept–concept interactions within each frame. It indicates which concepts influence a given concept and highlights additional concepts that are relevant at that moment. This provides a structured view of how concepts relate to each other spatially.

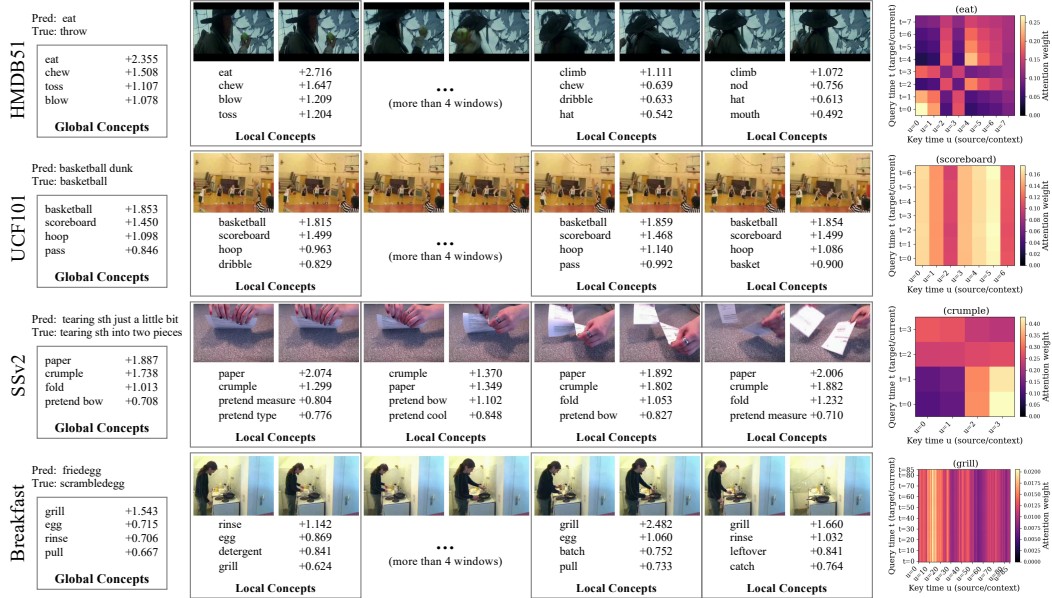

Figure 7: **MoTIF-ST explanations.** Correctly classified examples from Breakfast and UCF101, illustrating MoTIF with a space–time transformer (ViT-L/14) and the corresponding temporal and spatial attention matrices.

- The temporal attention block captures when a concept becomes important. It identifies the time steps at which a specific concept contributes most to the final prediction, thereby revealing the temporal structure of the activity.

Figure 7 shows two examples from Breakfast and UCF101, including their attention matrices and the corresponding explanations.

# B ADDITIONAL EXPERIMENTS

## B.1 WHERE *motifs* HELP TO UNDERSTAND MISCLASSIFICATIONS

In Figure 8, we illustrate representative failure cases for each dataset using models trained with ViT-L/14. All corresponding videos, including the full set of temporal concepts across all windows, are provided in the supplementary material.

Figure 8: **MoTIF explanations.** Example videos from all datasets with incorrect classifications.

The first example, from *Pirates of the Caribbean*, shows Barbossa eating an apple before throwing it away. Our model predicts *eat*, while the ground truth is *throw*. Global and local concept attributions reveal that the concept *eat*, triggered by the apple, was most strongly activated. The corresponding attention map illustrates that early query frames attend to early key frames, indicating that the model

anchors the decision on the moment where the apple is clearly visible and eaten. This concentrated attention explains why the model emphasizes *eat* over *throw*.

In the second example from UCF101, MoTIF detects correct concepts, such as *basketball* and *scoreboard*. However, these were insufficient to discriminate between the actions *basketball dunk* and *basketball*, leading to misclassification. Since the background frame remains nearly constant and only the players on the court are moving, the attention map for *scoreboard* shows a uniform distribution across time steps, with no clear temporal anchors. This indicates that the concept is consistently present.

The third example, from SSv2, highlights the dataset's inherent difficulty. Unlike the correctly classified case in Figure 2, MoTIF predicts *tearing sth just a little bit* instead of the ground truth *tearing sth into two pieces*. Concept activations focus primarily on hand movements, such as *crumple*, together with the *paper*. The attention map shows that the concept *crumple* receives strongest attention from the later key frames (u=2, u=3) across several query times while other frames receive lower, more diffuse weights.This diffuse attribution explains why MoTIF captures the general action but fails to resolve the fine-grained distinction required by SSv2.

Finally, in the Breakfast dataset, MoTIF correctly identifies and attends to concepts relevant for *egg*. Yet, the dataset contains two distinct egg-related actions, *friedegg* and *scrambledegg*, which leads to misclassification. Nevertheless, all identified concepts correspond to actions of a person cooking with eggs. Furthermore, the attention map for *grill* reveals a broad and diffuse distribution across many time steps, indicating that the concept is persistently active throughout the sequence rather than concentrated in a few decisive moments. This persistent but unspecific attention explains why MoTIF captures the general presence of egg-related cooking but fails to distinguish between the two fine-grained classes.

Although MoTIF does not always predict the correct class, its structure makes the reasoning process transparent by decomposing decisions into concepts and their temporal interactions. In this work, our emphasis was on the architecture rather than on designing the most suitable concept sets. We expect that future advances in concept extraction for video data will further improve performance, complementing MoTIF's interpretability with stronger predictive accuracy.

## B.2 DIAGONAL VS. FULL ATTENTION

We illustrate why full attention produces non-interpretable results by revisiting previously shown examples. In Figure 1, replacing diagonal attention with full attention shifts the most important concepts to *frown* (activation 4.796) and *face* (1.282), while all others fall below 0.01 and are omitted. Although these concepts appear as locally relevant, they do not correspond to the depicted action (class *bow*). This indicates that concept mixing occurs: full attention entangles channels, creates arbitrary concepts that are not visually apparent and thus undermine interpretability.

## B.3 INTERPRETING TEMPORAL DEPENDENCIES

An attention map visualizes how a concept channel distributes its focus across time. Each entry $W_{c,t,u}$ encodes the attention weight between query time step $t$ (vertical axis) and key/value time step $u$ (horizontal axis). A bright cell at position $(t, u)$ indicates that the activation of concept $c$ at time $t$ strongly attends to the representation of the same concept at time $u$. Diagonal patterns suggest that the concept mainly attends to itself at the same or nearby frames, while vertical stripes show that many query frames refer back to the same key frame, indicating the presence of a temporal anchor. In contrast, diffuse or uniform maps imply that the concept is expressed consistently across time rather than being tied to specific moments. Thus, by inspecting these maps, one can infer whether a concept is localized, persistent, or temporally linked to particular frames within the sequence.

## C  ADDITIONAL ABLATIONS

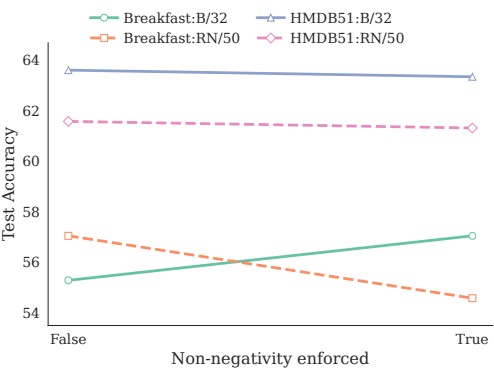 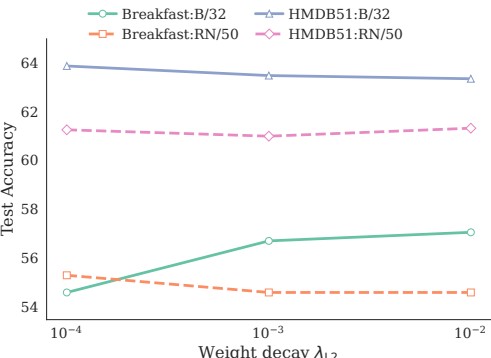

(a) **Non-negativity.** Test accuracy with and without enforcing non-negativity.

(b) **Weight decay.** Test accuracy across different weight decays.

Figure 9: **Architectural choices.** Effects of non-negativity and weight decay.

For completeness, we report further ablation studies that complement the main paper (Section 3.2.2).

**Nonnegativity.** The non-negativity constraint on classifier weights improves interpretability by ensuring class predictions are explained solely by positive concept evidence (Zhang et al., 2021). We enforce non-negativity of classifier weights $W$ by projection after each update. Results are shown in Figure 9a. This constraint tends to cause minor accuracy reductions for RN/50 on both datasets, while for B/32 on Breakfast we observe a slight improvement. Given the small fluctuations inherent in training, these effects should be interpreted as indicative rather than strictly conclusive. Overall, the effect on test accuracy is negligible. We therefore enable it for all experiments except SSv2, where modeling negative concepts is required to capture fine-grained actions.

**Weight decay.** Figure 9b reports test accuracy across different weight decay values for the AdamW optimizer. Performance remains largely stable, similar to the non-negativity constraint, except for Breakfast using B/32. We therefore fix the weight decay to $10^{-2}$ for all experiments.

**Per-concept affine transformation.** We optionally insert a per-concept affine transformation between the *per-channel temporal self-attention* and the classification head. While not strictly required, this block sharpens activations by rescaling and shifting each concept dimension before the nonlinearity. Figure 10 shows the effect on Breakfast and HMDB51: enabling the affine transformation yields a slight increase in test accuracy and a consistent reduction in both logits and concept entropy, indicating sharper and more decisive concept activations, especially on the Breakfast dataset.

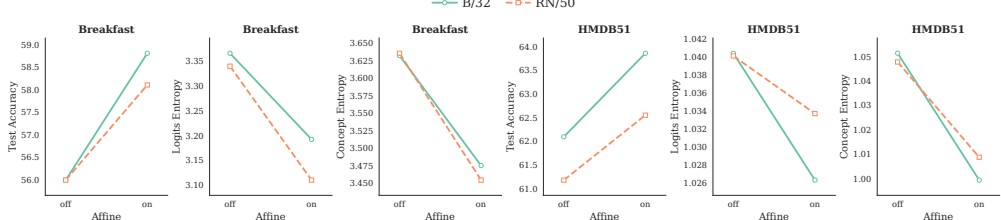

Figure 10: **Effect of the per-concept affine transformation.** Accuracy improves marginally, while entropy in both logits and concept activations decreases, suggesting that the affine block stabilizes and sharpens the CBM's internal representations.

**Classifier sparsity.** The sparsity penalty on classifier weights has a pronounced impact on test accuracy, as shown in Figure 11a. Larger values of $\lambda_{\ell_1}$ consistently reduce accuracy. We therefore set $\lambda_{\ell_1} = 10^{-3}$ in most experiments, as it offers a reasonable trade-off between regularization and performance.

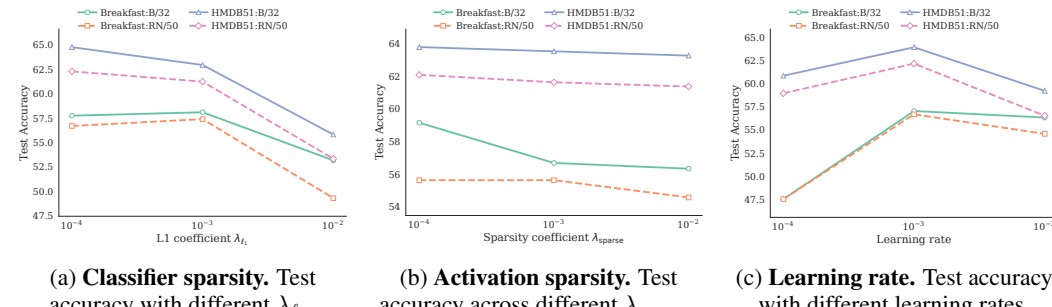

(a) **Classifier sparsity.** Test accuracy with different $\lambda_{\ell_1}$.

(b) **Activation sparsity.** Test accuracy across different $\lambda_{\text{sparse}}$.

(c) **Learning rate.** Test accuracy with different learning rates.

Figure 11: **Architectural choices.** Effects of classifier sparsity, activation sparsity, and learning rate.

**Activation sparsity.** The activation sparsity penalty shows little variation in test accuracy (see Figure 11b) across different values of $\lambda_{\text{sparse}}$, except for Breakfast. We attribute this to the comparatively long video sequences in that dataset. Accordingly, we use $\lambda_{\text{sparse}} = 10^{-3}$ for all experiments, except for Breakfast, where we set it to $10^{-4}$.

**Learning rate.** The learning rate has a strong effect on test accuracy, as shown in Figure 11c. For Breakfast and HMDB51, we set it to $10^{-3}$, while for UCF101 and SSv2, we use $10^{-4}$, which yielded better performance, an effect was not seen in these ablations.

**Window size.** We evaluate MoTIF with varying temporal input lengths to study robustness to sequence duration and efficiency trade-offs. While increasing the number of frames provides more temporal context, it also raises memory requirements and may not yield consistent accuracy gains. For comparability, we fix the batch size to 8 across all ablations, ensuring that changes in performance are solely attributable to window size rather than training dynamics. The results in Fig. 12 highlight that optimal window size depends on both dataset characteristics and backbone choice.

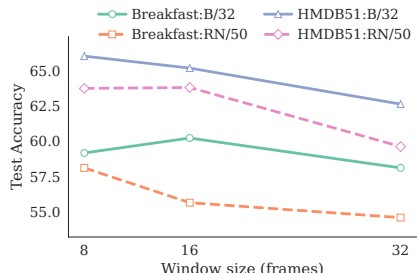

Figure 12: **Window size influence.** Test accuracy across different window sizes.

**Random seed.** Since most experiments were run with a fixed seed of 42 for reproducibility, we additionally ablate the effect of varying the random seed (39,40,41,42,43) for both MoTIF and the global CBM. As Figure 13 illustrates, the influence of the seed depends on the dataset. For HMDB51 the effect is negligible, whereas for Breakfast — particularly with the RN/50 backbone — we observe noticeably higher variance. Nevertheless, MoTIF consistently outperforms the global CBM. Moreover, the figure indicates that the reported numbers in Table 1 are conservative: even higher scores are achievable, but we deliberately refrain from reporting maxima and instead provide a representative overview.

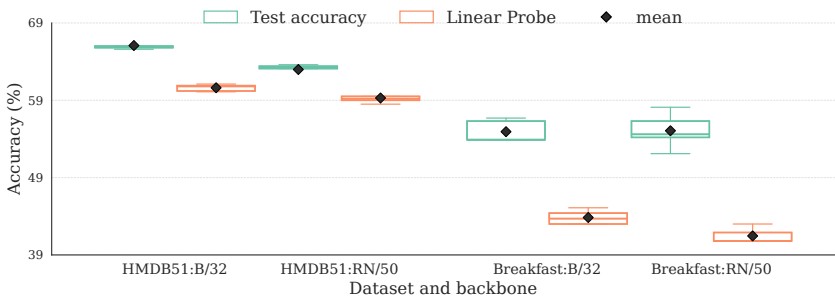

Figure 13: **Random seed.** Box plot of ablated datasets with five different seeds.

Additionally, since embeddings depend on the random choice of a representative image per window (previously fixed at seed 42), we re-embedded each dataset with five seeds. For each embedding variant we trained and evaluated MoTIF, the global CBM , and the zero-shot runs using a fixed training seed of 42 to isolate the effect of image-selection randomness. Table 4 reports the mean and standard deviation across embeddings (seeds 39–43). Std values are small overall; the largest observed variability is for Breakfast with RN/50 (std = 1.1).

Table 4: **Random seed on embeddings.** Mean and standard deviation (%) for methods on HMDB51 and Breakfast with two backbones.

| Dataset (backbone) | Method | Mean (%) | Std (%) |
|---|---|---|---|
| HMDB51 (RN/50) | MoTIF | 63.8 | 0.3 |
| | Global CBM | 60.2 | 0.3 |
| | Zero Shot | 29.4 | 0.2 |
| HMDB51 (B32) | MoTIF | 66.5 | 0.3 |
| | Global CBM | 61.0 | 0.6 |
| | Zero Shot | 38.7 | 0.3 |
| Breakfast (RN/50) | MoTIF | 53.2 | 1.1 |
| | Global CBM | 41.1 | 1.1 |
| | Zero Shot | 18.1 | 0.7 |
| Breakfast (B32) | MoTIF | 58.3 | 0.8 |
| | Global CBM | 43.9 | 0.6 |
| | Zero Shot | 25.8 | 0.2 |

---

[1]from (Liu et al., 2021) with eight clusters and all frames (cumulative)

## C.1 ACCURACIES TESTSPLITS

Tables 5–7 report Top-1 accuracies (%) on the test splits. Mean and standard deviation are computed across the shown splits (Breakfast: 4 splits; HMDB51 and UCF: 3 splits). For each backbone we list the MoTIF result (bold), the corresponding global CBM, and the zero-shot baseline. The reported standard deviation quantifies variability between test splits.

Table 5: **Top-1 accuracies (%) on Breakfast test splits.** Mean and standard deviation across splits.

| Model | s1 | s2 | s3 | s4 | Mean | Std |
|---|---|---|---|---|---|---|
| **CLIP RN/50 (MoTIF)** | 55.6 | 47.4 | 47.0 | 61.2 | 52.8 | 6.9 |
| Global CBM (RN/50) | 43.3 | 25.7 | 32.5 | 44.4 | 36.5 | 9.0 |
| Zero-shot (RN/50) | 17.3 | 19.2 | 16.0 | 21.9 | 18.6 | 2.6 |
| **CLIP B/32 (MoTIF)** | 56.3 | 44.9 | 51.3 | 61.0 | 53.4 | 6.9 |
| Global CBM (B/32) | 42.6 | 27.5 | 31.6 | 47.0 | 37.2 | 9.1 |
| Zero-shot (B/32) | 24.6 | 19.8 | 22.0 | 26.4 | 23.2 | 2.9 |
| **CLIP L/14 (MoTIF)** | 71.5 | 62.3 | 66.7 | 76.8 | 69.3 | 6.2 |
| **CLIP L/14 (MoTIF-ST)** | 73.2 | 64.8 | 65.0 | 90.8 | 73.5 | 12.2 |
| Global CBM (L/14) | 61.3 | 44.7 | 48.7 | 66.4 | 55.3 | 10.2 |
| Zero-shot (L/14) | 32.4 | 28.3 | 26.5 | 37.0 | 31.1 | 4.7 |
| **SigLIP L/14 (MoTIF)** | 76.1 | 62.1 | 73.1 | 82.7 | 73.5 | 8.6 |
| Global CBM (SigLIP L/14) | 59.9 | 44.7 | 53.3 | 70.6 | 57.1 | 10.9 |
| Zero-shot (SigLIP L/14) | 28.9 | 18.0 | 20.9 | 26.6 | 23.6 | 5.0 |
| **PE L/14 (MoTIF)** | 87.3 | 74.7 | 83.1 | 89.4 | 83.6 | 6.5 |
| **PE L/14 (MoTIF-ST)** | 86.2 | 74.7 | 82.3 | 91.2 | 83.6 | 7.0 |
| Global CBM (PE L/14) | 81.0 | 58.7 | 72.0 | 79.9 | 72.9 | 10.3 |
| Zero-shot (PE L/14) | 40.5 | 36.6 | 36.8 | 51.6 | 41.4 | 7.0 |

Table 6: **Top-1 accuracies (%) on HMDB51 test splits.** Mean and standard deviation across splits.

| Model | s1 | s2 | s3 | Mean | Std |
|---|---|---|---|---|---|
| **CLIP RN/50 (MoTIF)** | 64.1 | 62.3 | 62.1 | 62.8 | 1.1 |
| Global CBM (RN/50) | 58.6 | 60.1 | 59.2 | 59.3 | 0.8 |
| Zero-shot (RN/50) | 29.3 | 30.1 | 30.1 | 29.8 | 0.5 |
| **CLIP B/32 (MoTIF)** | 65.9 | 66.8 | 63.3 | 65.3 | 1.8 |
| Global CBM (B/32) | 62.0 | 63.0 | 59.8 | 61.6 | 1.6 |
| Zero-shot (B/32) | 38.4 | 37.9 | 38.0 | 38.1 | 0.3 |
| **CLIP L/14 (MoTIF)** | 73.8 | 73.9 | 72.2 | 73.3 | 1.0 |
| **CLIP L/14 (MoTIF-ST)** | 75.5 | 75.3 | 73.6 | 74.8 | 1.0 |
| Global CBM (L/14) | 68.5 | 68.8 | 67.9 | 68.4 | 0.5 |
| Zero-shot (L/14) | 45.8 | 45.6 | 45.6 | 45.7 | 0.1 |
| **SigLIP L/14 (MoTIF)** | 74.8 | 74.4 | 70.4 | 73.2 | 2.4 |
| Global CBM (SigLIP L/14) | 66.3 | 66.0 | 62.6 | 65.0 | 2.1 |
| Zero-shot (SigLIP L/14) | 48.4 | 50.0 | 49.5 | 49.3 | 0.8 |
| **PE L/14 (MoTIF)** | 79.9 | 79.3 | 79.6 | 79.6 | 0.3 |
| **PE L/14 (MoTIF-ST)** | 79.9 | 80.0 | 78.8 | 79.6 | 0.7 |
| Global CBM (PE L/14) | 74.0 | 75.0 | 74.1 | 74.4 | 0.6 |
| Zero-shot (PE L/14) | 56.7 | 57.3 | 56.2 | 56.7 | 0.6 |

Table 7: **Top-1 accuracies (%) on UCF test splits.** Mean and standard deviation across splits.

| Model | s1 | s2 | s3 | Mean | Std |
|---|---|---|---|---|---|
| **CLIP RN/50 (MoTIF)** | 82.4 | 82.5 | 83.4 | 82.8 | 0.6 |
| Global CBM (RN/50) | 80.7 | 80.0 | 79.3 | 80.0 | 0.7 |
| Zero-shot (RN/50) | 56.5 | 56.9 | 58.3 | 57.2 | 0.9 |
| **CLIP B/32 (MoTIF)** | 84.4 | 85.7 | 86.7 | 85.6 | 1.2 |
| Global CBM (B/32) | 82.2 | 82.5 | 83.6 | 82.8 | 0.7 |
| Zero-shot (B/32) | 59.4 | 60.2 | 60.1 | 59.9 | 0.4 |
| **CLIP L/14 (MoTIF)** | 92.4 | 93.7 | 93.6 | 93.2 | 0.7 |
| **CLIP L/14 (MoTIF-ST)** | 92.9 | 94.4 | 94.0 | 93.8 | 0.8 |
| Global CBM (L/14) | 88.8 | 91.0 | 90.3 | 90.0 | 1.1 |
| Zero-shot (L/14) | 71.1 | 70.6 | 70.1 | 70.6 | 0.5 |
| **SigLIP L/14 (MoTIF)** | 93.3 | 93.8 | 94.9 | 94.0 | 0.8 |
| Global CBM (SigLIP L/14) | 90.0 | 91.0 | 90.5 | 90.5 | 0.5 |
| Zero-shot (SigLIP L/14) | 80.0 | 81.9 | 79.2 | 80.4 | 1.4 |
| **PE L/14 (MoTIF)** | 94.6 | 95.7 | 95.8 | 95.4 | 0.7 |
| **PE L/14 (MoTIF-ST)** | 95.7 | 96.4 | 96.9 | 96.3 | 0.6 |
| Global CBM (PE L/14) | 94.6 | 93.9 | 95.0 | 94.5 | 0.6 |
| Zero-shot (PE L/14) | 73.8 | 75.6 | 74.4 | 74.6 | 0.9 |

# D   ALGORITHMS

This section summarizes MoTIF's procedures for training, test-time inference, and explanation. We adopt the notation from the main text: per-window concept activations $Z_{t,c}$, per-time logits $\ell_t$, and log-sum-exp (LSE) pooling with temperature $\tau$ and optional mask $m_t$.

---

**Algorithm 1** Training MoTIF (Moving Temporal Interpretable Framework)

---

1: **procedure** TRAIN($\{(X^{(n)}, y^{(n)})\}_{n=1}^N$, concept bank $\mathcal{C}$)
2:     Initialize Transformer parameters, affine $(\gamma, \delta)$, classifier $(W, b)$
3:     **for** each epoch **do**
4:         **for** each batch $(x, y)$ **do**
5:             **Temporal modeling:** per-channel temporal self-attention, to get $X_{t,c}^{(L)}$
6:             **Affine & nonnegativity:** $Z_{t,c} \leftarrow \text{Softplus}(\gamma_c X_{t,c}^{(L)} + \delta_c)$
7:             **Classification:** $\ell_t \leftarrow W Z_{t,:} + b$
8:             **Pooling:** $\hat{\ell} \leftarrow \text{LSE}_\tau(\{\ell_t\}, m)$
9:             **Loss:** $\mathcal{L} \leftarrow \text{CE}(\hat{\ell}, y) + \lambda_{\ell_1}\|W\|_1 + \lambda_{\text{sparse}} \frac{1}{(\sum_t m_t)C} \sum_{t,c} m_t |Z_{t,c}|$
10:            Update all parameters with AdamW
11:            **(Optional)** enforce nonnegativity: $W \leftarrow \max(W, 0)$
12:        **end for**
13:    **end for**
14:    **return** trained MoTIF
15: **end procedure**

---

**Description.** The training loop builds concept activations from window embeddings, refines them with per-channel temporal attention, applies a nonnegative affine projection, and classifies with a linear head pooled over time via LSE. The objective combines cross-entropy with sparsity regularizers; $W$ can be projected to enforce nonnegativity.

**Description.** Inference is a single forward pass: per-window logits are pooled with LSE to produce video-level logits, whose argmax yields the prediction.

---

**Algorithm 2** Inference with MoTIF (test-time forward pass)

---

1: **procedure** INFER(video $x$, trained MoTIF)
2:     Compute concept activations from window embeddings
3:     Apply per-channel temporal attention; affine + Softplus $\rightarrow$ nonnegative $Z_{t,c}$
4:     Compute per-time logits $\ell_t = W_k Z_{t,:} + b$
5:     Aggregate with LSE pooling: $\hat{\ell} = \text{LSE}_\tau(\{\ell_t\}, m)$
6:     Predict $y^* = \arg\max_k \hat{\ell}_k$
7:     **return** predicted label $y^*$
8: **end procedure**

---

**Algorithm 3** Explanation with MoTIF (global, local, temporal views)

---

1: **procedure** EXPLAIN(video $x$, class $k$, trained MoTIF)
2:     Run forward pass to obtain $Z_{t,c}$, $\ell_t$, and $\hat{\ell}$
3:     **Per-time contributions:** $\mathbf{c}_t^{(k)} \leftarrow Z_{t,:} \cdot W_{k,:}$;    $s_t^{(k)} \leftarrow \sum_c c_{t,c}^{(k)} + b_k$
4:     **Temporal importance:** $\pi_t^{(k)} \leftarrow \text{softmax}\big(s_t^{(k)}/\tau\big)$ over valid $t$ (mask $m_t$)
5:     **Global attribution:** $\bar{\mathbf{c}}^{(k)} \leftarrow \sum_t \pi_t^{(k)} \mathbf{c}_t^{(k)}$
6:     **Outputs:** $\bar{\mathbf{c}}^{(k)}$ (global concepts), top-$t$ by $\pi_t^{(k)}$ (local concepts), per-concept attention maps (temporal dependencies)
7:     **return** $\big(\bar{\mathbf{c}}^{(k)}, \{\pi_t^{(k)}\}_t, \text{attention maps}\big)$
8: **end procedure**

---

**Description.** Explanations decompose the prediction into per-concept contributions and reweight them by a time-importance distribution that mirrors LSE pooling. This yields three complementary views: (1) global concepts via $\bar{\mathbf{c}}^{(k)}$, (2) local concepts at decisive windows (large $\pi_t^{(k)}$), and (3) temporal dependencies from per-concept attention maps.

# E  CONCEPT SETS

## E.1  USED CONCEPTS FOR DATASETS

In Table 8, we show the number and kind of concepts used for the construction of MoTIF for each dataset. The number and kind of concepts vary for each dataset, since we asked the LLM to create domain-specific concepts that are useful for the downstream classification task.

Table 8: **MoTIF concepts.** The textual concepts utilized for all experiments which are listed in this paper.

| Set | Count | Concepts |
| --- | --- | --- |
| Breakfast | 223 | add, adjust, apple, arrange, assemble, avocado, bacon, bagel, bake, balance, banana, batter, beat, bin, blend, blender, blow, boil, bottle, bowl, bread, brew, brush, burner, butter, button, carry, carton, catch, cereal, chair, cheese, chop, cinnamon, clap, close, coffee, colander, comb, container, cook, cookbook, cool, core, counter, cover, crack, croissant, cucumber, cup, cupboard, cut, cuttingboard, detergent, dish, drag, drain, drizzle, drop, dry, egg, faucet, fill, flame, flip, fold, fork, freezer, fridge, froth, frown, fruit, fry, garbage, gesture, granola, grate, grater, grill, grind, ham, handle, heat, herb, hide, honey, hood, ice, ingredient, insert, jar, juice, kettle, knife, knob, knock, ladle, laugh, leftover, lid, mash, measure, measuringcup, measuringspoon, milk, mix, mug, napkin, nod, onion, open, orange, oven, ovenmitt, pack, package, pan, pantry, pastry, peel, peeler, pick, pinch, pit, place, plate, plug, point, poke, pour, preheat, press, pull, push, put, reach, recipe, recycle, release, remove, reveal, rinse, roll, rotate, sausage, scale, scoop, scramble, scrub, seal, serve, serving, set, shake, shave, sieve, sink, sip, sit, slice, slide, smile, snap, soap, socket, sort, spatula, spin, sponge, spoon, spread, sprinkle, squeeze, stack, stand, start, steam, steep, stir, stirrer, stool, stop, stove, strawberry, sugar, switch, syrup, table, take, tamp, tap, taste, tea, thermometer, throw, tie, tilt, timer, toast, tomato, tongs, toss, towel, tray, turn, twist, uncover, unfold, unscrew, unstack, untie, unwrap, warm, wash, waste, water, wave, whisk, wipe, wring, yogurt, zest, zip |
| UCF101 | 166 | aim, archer, archery, arena, arrow, athlete, balance, ball, bar, barbell, baseball, basket, basketball, bat, beam, bicycle, block, bounce, bow, bowl, boxing, breakdance, canoe, cap, catch, clap, climb, club, coach, control, court, cricket, curl, dance, dancer, deadlift, dismount, dive, dodge, dribble, dumbbell, enter, field, fight, flip, floor, frisbee, gallop, gloves, goal, goalpost, grab, grapple, grind, gun, gym, gymnast, handstand, hang, helmet, hit, hockey, hook, hoop, horse, hurdle, ice, instrument, jab, jersey, jump, kayak, kick, ladder, lane, lift, mat, microphone, mount, music, net, netting, opponent, pad, paddle, parry, pass, pedal, perform, pitch, platform, player, pool, press, puck, pull, push, racket, rail, raise, referee, reins, release, reload, ride, ring, rope, row, rower, rugby, run, sand, scoreboard, serve, sheet, shoot, shooter, sit, skateboard, skateboarder, skater, ski, skip, skis, smash, snow, snowboard, snowboarder, soccer, spike, spin, splash, sprint, squat, stadium, stage, stand, start, steer, stick, stop, strike, surf, surfboard, surfer, swim, swimmer, swing, sword, target, teammate, throw, timer, track, trampoline, tuck, turn, uniform, uppercut, volleyball, walk, wall, water, wave, wrestle, yoga |
| HMDB51 | 150 | apply, around, backward, balance, ball, baseball, basketball, bat, bend, bicycle, block, blow, bottle, bounce, bow, brake, brush, button, carry, cartwheel, catch, chair, chew, climb, close, comb, crawl, cross, crouch, cup, dismount, dive, door, down, drag, dribble, drink, drop, eat, enter, exit, face, fall, fight, finish, flip, float, frisbee, from, frown, gallop, grab, hair, hand, hands, handstand, hat, head, headstand, high, hit, hop, horse, hug, jacket, jog, juggle, jump, kick, kiss, knock, laugh, leap, left, leg, lie, lift, line, look, low, makeup, mount, mouth, nod, object, off, on, open, pedal, point, pull, punch, push, put, racket, reach, release, ride, right, roll, room, run, serve, shake, shave, shirt, shoelace, shoot, sing, sip, sit, skate, skateboard, ski, sled, sleep, slide, smile, snowboard, somersault, spin, sprint, stand, start, steer, stretch, surface, swim, swing, sword, take, talk, teeth, tennis, throw, tie, toss, touch, turn, untie, up, utensils, wake, walk, wash, wave, with, words, yawn, zip |
| SSv2 | 284 | accelerate, apple, arm, assemble, background, backpack, bag, balance, ball, banana, bend, bite, blow, book, bottle, bottom, bounce, bow, bowl, box, break, broken, can, cap, carrot, carry, catch, chair, chew, chop, chopstick, clap, clean, click, climb, close, closeable, cold, connect, container, cough, cover, crawl, crouch, crumple, cry, cucumber, cup, cut, dance, decelerate, dirty, disassemble, disconnect, door, downward, drag, drag mouse, draw, drink, drinkable, drop, dry, durable, eat, edible, empty, erase, face, fall, fasten, fill, finger, fixed, flatten, flip, floor, fold, fork, fragile, frown, fruit, full, gather, get up, grape, hand, heavy, hide, hold, hop, hot, insert, inside, juggle, jump, key, keyboard, kneel, knife, knock, laptop, laugh, lean, left, lid, lift, light, lock, loosen, mix, mouse, nod, object, open, openable, orange, other, outside, paint, paper, peel, pen, pencil, person, phone, plate, plug, point, pour, pourable, press, pretend to balance, pretend to block, pretend to bow, pretend to catch, pretend to catch fish, pretend to clap, pretend to clean, pretend to climb, pretend to close, pretend to cook, pretend to dance, pretend to dodge, pretend to draw, pretend to dribble, pretend to drink, pretend to drive, pretend to eat, pretend to fall, pretend to fire gun, pretend to honk, pretend to hug, pretend to jump rope, pretend to kick, pretend to kiss, pretend to load gun, pretend to lock, pretend to look around, pretend to measure, pretend to open, pretend to paddle, pretend to paint, pretend to play drums, pretend to play guitar, pretend to play piano, pretend to point, pretend to pour, pretend to pull, pretend to punch, pretend to push, pretend to read, pretend to row, pretend to salute, pretend to scroll, pretend to search, pretend to serve, pretend to shake hands, pretend to shoot arrow, pretend to shoot basket, pretend to sing, pretend to sleep, pretend to steer, pretend to steer wheel, pretend to stir, pretend to swing bat, pretend to swipe, pretend to throw, pretend to throw ball, pretend to type, pretend to unlock, pretend to use controller, pretend to wake, pretend to wave, pretend to weigh, pretend to write, pull, push, remote, remove, reveal, right, roll, rollable, rotate, rough, run, scatter, scoop, scroll, separate, shake, shake head, shelf, shout, sip, sit, sleep, slice, slide, smell, smile, smooth, snap, sneeze, speak, spill, spillable, spin, spin dance, spit, spoon, sprinkle, sprint, squeezable, stack, stackable, stand, start, stir, stop, stretch, stumble, surface, swing, swipe, table, tap, taste, tear, throw, tie, tighten, tilt, tomato, top, topple, touch, toy, turn off, turn on, type, uncover, unfasten, unfold, unlock, unplug, unstack, untie, unwrap, upward, vegetable, wake, walk, wall, wave, wet, whisper, window, wrap, write, yawn, zoom in, zoom out |

## E.2  CONCEPT SET VARIANCE

Table 9 lists the textual concepts used for the ablation in Figure 4. The sets differ because we prompted the LLM five times to generate concepts for the CBM using the following prompt:

```
Create unique concepts (>100) for a concept-bottleneck model for
the dataset 'Breakfast Actions'. Return them in this format:
"prepare coffee, grind beans, ..."
```

Both the number and the variety of concepts vary across calls. Despite this variability, model performance remains stable, demonstrating MoTIF's robustness to different concept sets. The same

prompting procedure was repeated for the other datasets (e.g. HMDB51, UCF101, Something-Something V2).

Table 9: **MoTIF Breakfast Concepts.** The textual concepts utilized for the ablation of concept influence.

| Set | Count | Concepts |
|---|---|---|
| Breakfast Original | 223 | add, adjust, apple, arrange, assemble, avocado, bacon, bagel, bake, balance, banana, batter, beat, bin, blend, blender, blow, boil, bottle, bowl, bread, brew, brush, burner, butter, button, carry, carton, catch, cereal, chair, cheese, chop, cinnamon, clap, close, coffee, colander, comb, container, cook, cookbook, cool, core, counter, cover, crack, croissant, cucumber, cup, cupboard, cut, cuttingboard, detergent, dish, drag, drain, drizzle, drop, dry, egg, faucet, fill, flame, flip, fold, fork, freezer, fridge, froth, frown, fruit, fry, garbage, gesture, granola, grate, grater, grill, grind, ham, handle, heat, herb, hide, honey, hood, ice, ingredient, insert, jar, juice, kettle, knife, knob, knock, ladle, laugh, leftover, lid, mash, measure, measuringcup, measuringspoon, milk, mix, mug, napkin, nod, onion, open, orange, oven, ovenmitt, pack, package, pan, pantry, pastry, peel, peeler, pick, pinch, pit, place, plate, plug, point, poke, pour, preheat, press, pull, push, put, reach, recipe, recycle, release, remove, reveal, rinse, roll, rotate, sausage, scale, scoop, scramble, scrub, seal, serve, serving, set, shake, shave, sieve, sink, sip, sit, slice, slide, smile, snap, soap, socket, sort, spatula, spin, sponge, spoon, spread, sprinkle, squeeze, stack, stand, start, steam, steep, stir, stirrer, stool, stop, stove, strawberry, sugar, switch, syrup, table, take, tamp, tap, taste, tea, thermometer, throw, tie, tilt, timer, toast, tomato, tongs, toss, towel, tray, turn, twist, uncover, unfold, unscrew, unstack, untie, unwrap, warm, wash, waste, water, wave, whisk, wipe, wring, yogurt, zest, zip |
| Breakfast Set 2 | 140 | adjust heat, arrange cutlery, bake bread, bake pastry, beat eggs, bend down, bite food, break chocolate, break egg shell, breakfast clock, bubbling liquid, butter toast, chair at table, chew food, chop onion, clean spoon, close carton, close cupboard, close drawer, close fridge, close jar, close microwave, close oven, close tap, cut banana, cut dough, cut sandwich, dice tomato, drip water, drizzle dressing, drizzle honey, drizzle oil, dry dish, dry hands, empty sink, family sitting, flip bread, flip toast, fold mixture, fold napkin, grab fork, grab knife, grab spoon, grate chocolate, grill sandwich, gulp drink, hold bowl, hold glass, hold plate, hold straw, knead dough, lean forward, lick spoon, mash egg, mash potato, melt chocolate, mix salad, mop spill, morning light, move chair, one person eating, open carton, open cupboard, open drawer, open fridge, open microwave, open oven, open tap, peel apple, peel banana, peel egg, peel orange, person standing at stove, place utensil, pour batter, pour carton, preheat oven, press button, put down bowl, put down glass, put down plate, reach cupboard, reach shelf, recycle carton, recycle glass, recycle paper, recycle plastic, rest dough, rinse cup, roast vegetable, roll dough, run water, separate yolk, set napkin, shape dough, shred lettuce, sip drink, sip straw, sit down, sizzling pan, slice cucumber, soap hands, spread batter, spreading butter, spreading jam, spreading topping, sprinkle cheese, sprinkle herbs, sprinkle spices, squeeze lemon, stack plates, stand up from chair, steam rising, stir chocolate, swallow food, take carton, take jar, take package, take utensil, tear package, throw trash, toast bun, toss salad, towel hands, turn knob, turn off kettle, turn off stove, turn on kettle, turn on stove, two people cooking, unwrap sandwich, wash dish, wash hands, water boiling, whisk whites, wipe counter, wipe knife, wipe plate, wrap sandwich, zest lemon |
| Breakfast Set 3 | 128 | add cinnamon, add fruit topping, add granola, add honey, add ice to blender, add milk, add milk to cereal, add sugar, arrange cutlery, assemble sandwich, bake pastry, beat eggs, blend smoothie, boil potato, boil water, brew coffee, butter toast, check timer, chop herbs, chop onion, chop vegetables, clear table, close cupboard, close fridge, close jar, close oven, cook bacon, cook pancake, cook sausage, core apple, crack egg, cut sandwich, dice vegetables, drain bacon, drizzle honey, drizzle syrup, dry dishes, fill kettle, flip bacon, flip omelette, flip pancake, follow recipe, froth milk, fry egg, grate cheese, grill sandwich, grind coffee beans, heat pan, insert coffee pod, make omelette, mash avocado, mash potato, measure ingredients, mix batter, open cupboard, open egg carton, open fridge, open jar, open oven, operate espresso machine, pack leftovers, peel banana, peel potato, pick up spoon, pit avocado, place cup, place plate, pour batter, pour cereal into bowl, pour coffee into cup, pour hot water, pour milk, pour smoothie, pour syrup, pour yogurt into bowl, preheat oven, prepare coffee, pull espresso shot, put ingredient in fridge, put leftovers in fridge, read recipe, rinse fruit, scramble eggs, serve pancakes, set table, set timer, sip beverage, slice apple, slice avocado, slice bagel, slice banana, slice bread, slice cheese, slice cucumber, slice ham, slice orange, slice strawberries, slice tomato, spread butter, spread cream cheese, spread jam, spread peanut butter, sprinkle sugar, squeeze lemon, steam milk, steep tea, stir beverage, strain smoothie, take cup, take ingredient from fridge, take leftovers out fridge, take plate, toast bagel, toast bread, unscrew lid, unwrap bread, use fork, use knife, use measuring cup, use measuring spoon, use spatula, use tongs, use whisk, warm croissant, wash dishes, wash fruit, whisk eggs, wipe counter |
| Breakfast Set 4 | 175 | add cereal, add cinnamon, add cocoa powder, add honey, add ice to blender, add milk, add milk to coffee, add pepper, add salt, add sugar to coffee, add sugar to tea, add toppings, adjust seasoning, arrange cutlery, assemble sandwich, beat eggs, blend smoothie, blow on hot food, boil potato, boil water, brew coffee, butter toast, carry plate to table, check food temperature, check timer, chop herbs, chop onion, chop tomato, clean blender, clean counter, clear table, close cupboard, close egg carton, close fridge, close jar, close microwave, close milk carton, close oven, cook bacon, cook sausage, crack egg, cut lemon, cut sandwich, dice vegetables, drain bacon, drizzle honey, drizzle syrup, dry dishes, dry hands, fill kettle, fill pot with water, flip bacon, flip omelette, flip pancake, follow recipe, froth milk, fry egg, grate cheese, grind coffee beans, hold pan lid, insert bread into toaster, insert coffee pod, make omelette, mash potato, measure ingredients, mix batter, open cupboard, open egg carton, open fridge, open jar, open microwave, open milk carton, open oven, open package, operate blender, operate espresso machine, pack leftovers, peel banana, peel orange, peel potato, pick up cup, pick up knife, pit avocado, place pan off stove, place pan on stove, place plate, pour batter, pour cereal into bowl, pour coffee, pour eggs into pan, pour from carton, pour hot water, pour milk into bowl, pour pancake batter, pour smoothie, pour syrup, pour tea, pour yogurt into bowl, preheat oven, prepare coffee, prepare tea, press coffee, pull espresso shot, put leftovers in fridge, reach for ingredient, read recipe, remove bread from toaster, remove lid from pot, rinse fruit, scoop butter, scramble eggs, search for ingredient, season food, serve omelette, serve pancakes, set table, set timer, sip beverage, sit down, slice apple, slice avocado, slice bagel, slice banana, slice bread, slice cheese, slice cucumber, slice fruit, slice ham, slice kiwi, slice orange, slice pancake stack, slice strawberries, slice tomato, spoon yogurt, spread butter, spread cream cheese, spread jam, spread peanut butter, sprinkle granola, sprinkle sugar, squeeze lemon, start microwave, steam milk, steep tea, stir coffee, stop microwave, strain smoothie, take leftovers out fridge, take plate, taste food, toast bagel, toast bread, turn off kettle, turn off stove, turn on kettle, turn on stove, unscrew lid, unwrap bread, use fork, use french press, use knife, use measuring cup, use oven mitts, use spatula, use spoon, use toaster, use tongs, use whisk, wash blender, wash dishes, wash fruit, wash hands, whisk eggs, wipe counter |
| Breakfast Set 5 | 161 | add cinnamon, add fruit topping, add granola, add honey, add ice to blender, add milk, add milk to cereal, add nuts, add sugar, adjust seasoning, arrange cutlery, assemble sandwich, bake pastry, beat eggs, blend smoothie, blow on hot food, boil potato, boil water, brew coffee, butter toast, carry plate to table, check timer, chop herbs, chop onion, chop vegetables, clear table, close cupboard, close fridge, close jar, close milk carton, close oven, cook bacon, cook pancake, cook sausage, core apple, core pineapple, crack egg, cut parsley, cut pineapple, cut sandwich, dice vegetables, drain bacon, drain can, drizzle honey, drizzle syrup, dry dishes, dry hands, fill kettle, flip bacon, flip omelette, flip pancake, follow recipe, froth milk, fry egg, grate cheese, grill sandwich, grind beans, heat pan, insert coffee pod, juice orange, make omelette, mash avocado, mash potato, measure ingredients, mix batter, open cupboard, open egg carton, open fridge, open jar, open milk carton, open oven, open package, open tin, operate espresso machine, pack leftovers, peel banana, peel orange, peel potato, pick up cup, pick up spoon, pit avocado, place cup, place pan off stove, place plate, pour batter, pour cereal, pour coffee into cup, pour from carton, pour hot water, pour milk, pour smoothie, pour syrup, pour yogurt, preheat oven, prepare coffee, pull espresso shot, put ingredient in fridge, put leftovers in fridge, reach for ingredient, read recipe, remove lid from pot, rinse fruit, scramble eggs, seal container, serve pancakes, set table, set timer, sip beverage, sit down, slice apple, slice avocado, slice bagel, slice banana, slice bread, slice cheese, slice cucumber, slice ham, slice kiwi, slice lemon, slice melon, slice orange, slice pear, slice strawberries, slice tomato, spread butter, spread cream cheese, spread jam, spread peanut butter, sprinkle sugar, squeeze lemon, stand up, steam milk, steep tea, stir beverage, strain smoothie, take cup, take ingredient from fridge, take leftovers out fridge, take plate, taste food, toast bagel, toast bread, toast nuts, unscrew lid, unwrap bread, unwrap package, use fork, use knife, use measuring cup, use measuring spoon, use oven mitts, use spatula, use tongs, use whisk, warm croissant, wash dishes, wash fruit, wash hands, whisk eggs, wipe counter, zest lemon |

### E.3 ABLATION OF CONCEPT CONSTRUCTION

To evaluate whether MoTIF can effectively ground explicitly temporal concepts, we constructed five concept sets for SSv2, the dataset in our benchmark that relies most heavily on temporal reasoning: (1) nouns only, (2) verbs only, (3) nouns combined with verbs, (4) the curated concept set used in the main paper, and (5) the union of all concepts (see Table 10). All variants were evaluated using MoTIF with and without space-time transformer architecture. Across all settings, MoTIF consistently outperforms Global CBM.

Table 10: **MoTIF Breakfast Concepts.** Textual concepts used in the concept-set ablation.

| Set | Count | Concepts |
|---|---|---|
| **Nouns-Only Concepts** | 60 | object, container, box, cup, bowl, plate, spoon, knife, fork, chopstick, pen, pencil, paper, book, phone, remote, laptop, keyboard, mouse, bag, backpack, toy, ball, fruit, apple, orange, banana, grape, vegetable, carrot, cucumber, tomato, bottle, can, lid, cap, key, lock, door, window, wall, floor, table, chair, shelf, hand, finger, arm, face, person, background, surface, inside, outside, top, bottom, left, right, upward, downward |
| **Verbs-Only Concepts** | 135 | push, pull, lift, drop, hold, carry, throw, catch, slide, drag, roll, spin, rotate, flip, fold, unfold, wrap, unwrap, tie, untie, fasten, unfasten, tighten, loosen, break, cut, slice, chop, tear, peel, crumple, flatten, bend, stretch, shake, stir, pour, scoop, sprinkle, stack, unstack, assemble, disassemble, open, close, lock, unlock, press, tap, swipe, scroll, zoom in, zoom out, point, touch, wave, clap, knock, snap, swing, juggle, bounce, balance, topple, insert, remove, fill, empty, mix, separate, spill, scatter, gather, cover, uncover, hide, reveal, lean, tilt, climb, crawl, jump, hop, walk, run, sprint, stumble, fall, get up, sit, stand, kneel, crouch, bow, dance, nod, shake head, smile, frown, laugh, cry, shout, whisper, speak, yawn, sneeze, cough, sleep, wake, eat, chew, bite, sip, drink, spit, blow, smell, taste, write, draw, erase, paint, type, click, drag mouse, plug, unplug, connect, disconnect, turn on, turn off, start, stop, accelerate, decelerate |
| **Noun–Verb Concepts** | 104 | lift the box, open the box, close the box, drop the box, push the box, pull the box, carry the cup, pour from the cup, fill the cup, empty the cup, hold the cup, rotate the cup, open the bottle, close the bottle, pour from the bottle, lift the bottle, drink from the bottle, cut the paper, fold the paper, tear the paper, crumple the paper, write on the paper, open the book, close the book, flip the book, read the book, drop the book, type on the keyboard, plug in the laptop, unplug the laptop, open the laptop, close the laptop, click the mouse, drag the mouse, press the key, turn on the phone, turn off the phone, swipe the phone, tap the phone, scroll on the phone, charge the phone, unlock the phone, lock the door, unlock the door, open the door, close the door, lift the chair, move the chair, sit on the chair, stand from the chair, throw the ball, catch the ball, bounce the ball, roll the ball, peel the banana, eat the banana, cut the apple, eat the apple, slice the cucumber, pour the water, stir the soup, mix the ingredients, chop the vegetables, open the can, close the lid, place the lid, remove the lid, stack the boxes, unstack the boxes, wrap the gift, unwrap the gift, tie the rope, untie the rope, press the button, flip the switch, turn the knob, insert the plug, remove the plug, connect the cable, disconnect the cable, shake the bottle, squeeze the bottle, pour the juice, stir the drink, draw on the paper, paint the wall, erase the drawing, fold the towel, open the backpack, close the backpack, pick up the bag, drop the bag, open the window, close the window, wipe the table, clean the floor, open the box with one hand, lift the object, move the object, drop the object, hold the object, place the object, throw the object, pick up the object |
| **All Concepts** | 391 | push, pull, lift, drop, hold, carry, throw, catch, slide, drag, roll, spin, rotate, flip, fold, unfold, wrap, unwrap, tie, untie, fasten, unfasten, tighten, loosen, break, cut, slice, chop, tear, peel, crumple, flatten, bend, stretch, shake, stir, pour, scoop, sprinkle, stack, unstack, assemble, disassemble, open, close, lock, unlock, press, tap, swipe, scroll, zoom in, zoom out, point, touch, wave, clap, knock, snap, swing, juggle, bounce, balance, topple, insert, remove, fill, empty, mix, separate, spill, scatter, gather, cover, uncover, hide, reveal, lean, tilt, climb, crawl, jump, hop, walk, run, sprint, stumble, fall, get up, sit, stand, kneel, crouch, bow, dance, spin dance, nod, shake head, smile, frown, laugh, cry, shout, whisper, speak, yawn, sneeze, cough, sleep, wake, eat, chew, bite, sip, drink, spit, blow, smell, taste, write, draw, erase, paint, type, click, drag mouse, plug, unplug, connect, disconnect, turn on, turn off, start, stop, accelerate, decelerate, pretend to push, pretend to pull, pretend to pour, pretend to eat, pretend to drink, pretend to throw, pretend to catch, pretend to type, pretend to swipe, pretend to scroll, pretend to climb, pretend to fall, pretend to hug, pretend to kiss, pretend to wave, pretend to play guitar, pretend to drive, pretend to steer, pretend to read, pretend to sleep, pretend to wake, pretend to write, pretend to draw, pretend to paint, pretend to clean, pretend to cook, pretend to stir, pretend to measure, pretend to weigh, pretend to look around, pretend to search, pretend to point, pretend to balance, pretend to open, pretend to close, pretend to lock, pretend to unlock, pretend to kick, pretend to punch, pretend to block, pretend to dodge, pretend to jump rope, pretend to row, pretend to paddle, pretend to shoot arrow, pretend to load gun, pretend to fire gun, pretend to throw ball, pretend to dribble, pretend to shoot basket, pretend to swing bat, pretend to serve, pretend to catch fish, pretend to steer wheel, pretend to honk, pretend to use controller, pretend to play piano, pretend to play drums, pretend to dance, pretend to sing, pretend to clap, pretend to salute, pretend to bow, pretend to shake hands, pretend to hug, pretend to kiss, lift the box, open the box, close the box, drop the box, push the box, pull the box, carry the cup, pour from the cup, fill the cup, empty the cup, hold the cup, rotate the cup, open the bottle, close the bottle, pour from the bottle, lift the bottle, drink from the bottle, cut the paper, fold the paper, tear the paper, crumple the paper, write on the paper, open the book, close the book, flip the book, read the book, drop the book, type on the keyboard, plug in the laptop, unplug the laptop, open the laptop, close the laptop, click the mouse, drag the mouse, press the key, turn on the phone, turn off the phone, swipe the phone, tap the phone, scroll on the phone, charge the phone, unlock the phone, lock the door, unlock the door, open the door, close the door, lift the chair, move the chair, sit on the chair, stand from the chair, throw the ball, catch the ball, bounce the ball, roll the ball, peel the banana, eat the banana, cut the apple, eat the apple, slice the cucumber, pour the water, stir the soup, mix the ingredients, chop the vegetables, open the can, close the lid, place the lid, remove the lid, stack the boxes, unstack the boxes, wrap the gift, unwrap the gift, tie the rope, untie the rope, press the button, flip the switch, turn the knob, insert the plug, remove the plug, connect the cable, disconnect the cable, shake the bottle, squeeze the bottle, pour the juice, stir the drink, draw on the paper, paint the wall, erase the drawing, fold the towel, open the backpack, close the backpack, pick up the bag, drop the bag, open the window, close the window, wipe the table, clean the floor, open the box with one hand, lift the object, move the object, drop the object, hold the object, place the object, throw the object, pick up the object, object, container, box, cup, bowl, plate, spoon, knife, fork, chopstick, pen, pencil, paper, book, phone, remote, laptop, keyboard, mouse, bag, backpack, toy, ball, fruit, apple, orange, banana, grape, vegetable, carrot, cucumber, tomato, bottle, can, lid, cap, key, lock, door, window, wall, floor, table, chair, shelf, hand, finger, arm, face, person, other, background, surface, inside, outside, top, bottom, left, right, upward, downward, hot, cold, wet, dry, clean, dirty, empty, full, broken, fixed, smooth, rough, heavy, light, fragile, durable, rollable, stackable, squeezable, pourable, spillable, openable, closeable, edible, drinkable |

Table 11: **Concept-Set Ablation on SSv2.** Top-1 accuracy (%) with PE-L/14 evaluated on the four additional concept sets.

| Concept Set | MoTIF | MoTIF-ST | Global CBM |
|---|---|---|---|
| (1) Nouns only | 18.0 | 20.3 | 14.8 |
| (2) Verbs only | 24.4 | 28.6 | 21.2 |
| (3) Nouns + Verbs | 22.1 | 24.5 | 18.2 |
| (4) All concepts | 29.8 | 36.0 | 26.7 |
| (5) Original set | 30.0 | 35.1 | 25.5 |

The ablation yields four observations:

- Across all concept sets, MoTIF improves over Global CBM, showing that MoTIF's temporal modeling reliably strengthens concept grounding regardless of the chosen vocabulary.

- Verb-only concepts (set 2) achieve better performance than the noun-only (set 1) counterpart, indicating that MoTIF is particularly effective at grounding action dynamics that require temporal structure.

- The curated concept set (set 4) yields the highest performance (36.0% - with MoTIF-ST), suggesting that a balanced vocabulary provides the best trade-off between coverage and specificity.

