# OpenReview forum: "Concepts in Motion: Temporal Bottlenecks for Interpretable Video Classification"
_ICLR.cc/2026/Conference — Submitted to ICLR 2026_

### Official Review · Reviewer_y1P2 · 2025-10-24

**Soundness:** 3
**Presentation:** 3
**Contribution:** 2
**Rating:** 4
**Confidence:** 3

**Summary:**

This work presents MoTIF, an interpretable framework for video classification that adapts the concept bottleneck model (CBM) idea to temporal data. The method introduces a transformer-like architecture with per-concept temporal attention, enabling explanations at global, local, and temporal levels. The goal is to capture semantically meaningful concepts across frames and use them to explain and predict actions.

While the paper is well-motivated and technically clear, the approach faces representational constraints. By forcing all reasoning through a small set of predefined, frame-level concepts, MoTIF loses much of the expressive capacity that makes modern end2end, space-time video transformers successful. As a result, its significantly underperforms on challenging datasets such as Something-Something v2, where end-to-end models achieve far stronger results.

**Strengths:**

The paper is relatively well written and easy to follow.

The proposed approach is sound.

The resulting model offers intuitive explainability of its action classification predictions.

A reasonable ablation study is reported.

Experiments show that, at least for frame-level visual representations, adding a CBM head does not results in a drop in performance.

**Weaknesses:**

The key concerns I have regarding this work is that all the results are shown for frame-level visual representations that are fundamentally limited in their ability to capture spatiotemporal concepts in action classification. As a results, the strongest variant only achieves a classification accuracy of 30% on SSv2 (compared to 76% for the current sota method on this dataset). It is unclear if the proposed approach is applicable to modern space-time transformer architectures, which makes it hard to gauge its impact on the community. To address this issue, the authors need to report results with sota space-time transformer representations on non-toy datasets (e.g. on SSv2).

In the same vein, the sota methods' results need to be moved from the appendix to Table 1 to provide better context to the reader.

Furthermore, variance in performance form concept sampling should be reported for every backbone it Table 1 directly. This would likely make some improvements over linear probing baseline not statistically significant at least on some datasets.

As a final comment for Table 1, the authors need to report runtime for all the variants, as this is an important factor to consider.

Evidence of the fact that diagonal attention is not introducing performance penalties are not compelling, as only toy datasets (like hmdb) and weak architectures are ablated. It would be much stronger to show this on SSv2 with top-performing backbones.

**Questions:**

Please results with sota space-time transformer representations on non-toy datasets (e.g. on SSv2).

Please move sota results from the appendix to Table 1 and add runtime for all the variants as well as variance from concept sampling for your method.

Please show diagonal attention ablation on SSv2 with top-performing backbones

---

> ### Author Response · Authors · 2025-11-18
> **Rebuttal (1/2)**
>
> We thank the reviewer for the constructive feedback and the positive assessment of our work, particularly the comments that the paper is well written, the approach is sound, and the explanations are intuitive.
>
> Below, we address the reviewer’s specific concerns with new experiments and additional analyses, focusing in particular on integrating space-time transformers into MoTIF.
>
> First, a comment on SSv2 performance. Concurrent work that also applies CBMs to video data—but without temporal information [1] and accepted as a NeurIPS Spotlight—reports strong results on UCF101, which we outperform. However, they evaluate only on comparatively “easier” datasets. We intentionally included SSv2 despite its lower absolute performance because this gap is important for advancing concept-based interpretability on challenging fine-grained datasets. As recent work shows [2], video embedders struggle heavily with distinguishing chirality-sensitive actions. Our hypothesis is that temporal-aware representations will substantially improve MoTIF and related approaches. The field is fortunately moving in this direction.
>
> ---
>
> > [...] It is unclear if the proposed approach is applicable to modern space-time transformer architectures, which makes it hard to gauge its impact on the community. To address this issue, the authors need to report results with sota space-time transformer representations on non-toy datasets (e.g. on SSv2).
>
> We appreciate this suggestion. Although this was part of our planned future work, your comment motivated us to conduct these experiments during the rebuttal period. We are pleased to report that diagonal attention integrates seamlessly with space-time transformers, and the results support our claims.
>
> The experimental setting matches the previously described setup. Below in Table 1, we report results on all datasets using ViT-L/14 and PE-L/14.
>
>
> **Table 1: MoTIF with space-time tranformers**
>
> |Backbone |Breakfast|HMDB51|UCF101|SSv2|
> |-----|--------|--------|--------|--------|
> |MoTIF (L/14) |73.5 +/- 12.2       | 74.8 +/- 1.0       |93.8 +/- 0.9       |23.9       |
> |MoTIF (PE-L/14)  |83.6 +/- 7.0     |79.6 +/- 0.7       |96.3 +/- 0.6       |35.1       |
>
> The updated version includes code, results, and visual examples showing that concept isolation remains intact. These findings further strengthen our claims, and we reference them in multiple responses. We also ablate additional configurations, including TimesFormer, as requested. Thank you for enabling this important extension.
>
>
> > [...] It would be much stronger to show this on SSv2 with top-performing backbones. Please show diagonal attention ablation on SSv2 with top-performing backbones
>
> Figure 4 already shows that diagonal attention introduces penalties. We agree, however, that explicitly reporting SSv2 performance with SOTA backbones is valuable. In Table 2 are the results for the best-performing backbones, including their space-time variants.
>
> **Table 2: MoTIF space-time tranformer with and without diagonal attention**
>
> | Model | Accuracy | Epoch Time (s) |
> |-------------|---------------------|------------|
> | MoTIF-ViT-L/14 (diagonal)| 20.4% | 10.4 |
> | MoTIF-ViT-L/14 (full)| 27.8% | 10.1 |
> | MoTIF-ST-ViT-L/14 (diagonal)| 23.9% | 26.8 |
> | MoTIF-ST-ViT-L/14 (full)| 26.0% | 26.8 |
> | MoTIF-ViT-PE-L14 (diagonal)| 30.0% | 10.5 |
> | MoTIF-ViT-PE-L14 (full)| 40.1% | 10.0 |
> | MoTIF-ST-ViT-PE-L14 (diagonal)| 35.1% | 26.7 |
> | MoTIF-ST-ViT-PE-L14 (full)| 41.4% | 26.7 |

---

> > ### Author Response · Authors · 2025-11-18
> > **Rebuttal (2/2)**
> >
> > > Please move sota results from the appendix to Table 1 and add runtime for all the variants as well as variance from concept sampling for your method.
> >
> >
> > We follow the reviewer’s recommendation and moved the results to the main paper, including the addition of runtime measurements. However, adding the variance from concept sampling for every dataset and backbone configuration would, in our view, make the table difficult to read.
> > Nevertheless, in accordance with reviewer [qZ1Z], we additionally ablate the effect of different concept types on SSv2, as it is the most demanding dataset.
> >
> > Here we investigate whether MoTIF can meaningfully ground explicitly temporal concepts. For this, we created five concept sets for SSv2, which most relies on such temporal concepts: (1) nouns only, (2) verbs only, (3) nouns combined with verbs, (4) the original concept set used in our paper, and (5) all concepts together (see updated appendix for full concept lists and Table 3). We tested these with the updated TimesFormer architecture (addressed by reviewer [y1P2]) using MoTIF, which consistently outperforms Global CBM across all concept sets. For your information, the term linear probe and the way we originally described it caused confusion, so we have renamed this baseline to Global CBM.
> >
> > **Table 3: Effect of concept set on SSv2**
> >
> > | Concept Set | MoTIF |MoTIF-ST | Global CBM |
> > |-|-|-|-|
> > | (1) Nouns only | 18.0% |20.3% | 14.8% |
> > | (2) Verbs only | 24.4% |28.6% | 21.2% |
> > | (3) Nouns + Verbs | 22.1% |24.5% | 18.2% |
> > | (4) Original set | 30.0% |35.1% | 25.5% |
> > | (5) All concepts | 29.8% |36.0% | 26.7% |
> >
> > As expected from the CBM literature, the concept set has a significant impact on performance. For SSv2 in particular, a large and varied concept vocabulary is beneficial. If this experiment is not fully aligned with what the reviewer had in mind, we are prepared to run additional analyses during the rebuttal period. More details are provided in the updated appendix.
> >
> > [1] Disentangled Concepts Speak Louder Than Words: Explainable Video Action Recognition, NeurIPS 2025
> >
> > [2] Chirality in Action: Time-Aware Video Representation Learning by Latent Straightening, NeurIPS 2025
> >
> > ---
> >
> > We hope that our additional experiments and clarifications fully address your concerns. Your suggestions significantly strengthened the paper, and we would be very happy if the revised version meets your expectations.
> >
> > Best regards,
> > MoTIF authors

---

> > > ### Comment · Reviewer_y1P2 · 2025-11-24
> > >
> > > I thank the authors for their detailed response and appreciate the additional experiments, particularly those incorporating space–time transformer backbones. However, several points remain only partially addressed, and some of the new results reinforce rather than alleviate my initial concerns.
> > >
> > > 1. The results with space–time transformer backbones, while improved over 2D variants, remain roughly half of current SOTA performance. This supports the view that the proposed interpretable architecture imposes a substantial bottleneck on representational capacity.
> > >
> > > 2. Variance from concept sampling is not reported in the main SOTA comparison table. While I understand the authors’ concern about readability, these values should at least have been included in the rebuttal for transparency.
> > >
> > > 3. Runtime is provided only for the proposed approach, not for the baselines, which limits the interpretability of the comparison.
> > >
> > > 4. The additional ablations of diagonal attention confirm that it introduces significant performance penalties across architectures on SSv2.
> > >
> > > Overall, the rebuttal suggests that the interpretability offered by MoTIF comes at a considerable cost to performance in realistic settings. This limits the practical impact of the method, even though it retains some academic value as a step toward interpretable video models. I therefore intend to maintain my original score.

---

> > > > ### Author Response · Authors · 2025-11-26
> > > >
> > > > We thank the reviewer for carefully considering our rebuttal and for the additional feedback. Please find our responses to the remaining concerns below.
> > > >
> > > > ---
> > > >
> > > > > Performance gap to SOTA
> > > >
> > > > We agree that a substantial performance gap remains between current SOTA video classification models and our proposed interpretable architecture on SSv2. However, we do not consider this gap to be a weakness of our method. Instead, it reflects the broader challenges and limitations associated with the *current state of interpretable modeling*, rather than an inherent shortcoming of our approach.
> > > >
> > > > In our view, there are two distinct use cases: (i) models aiming for interpretability, and (ii) models optimized purely for performance. Our method is designed for the former, and within this space we demonstrate clear, significant improvements over existing interpretable approaches (e.g., compared to [1]). This contribution holds independently of comparisons to non-interpretable SOTA models.
> > > >
> > > > Additionally, recent work [4] has shown that current video embedders struggle considerably with chirality-sensitive actions. We hypothesize that future, more temporally aware representations will substantially improve MoTIF’s performance. Likewise, advances in concept extraction [2,3] are likely to narrow the gap to full-complexity SOTA methods.
> > > >
> > > > > Variance of concept sampling
> > > >
> > > > We agree and to ensure full transparency, we report results for L/14 and PE-L/14 using both MoTIF and MoTIF-ST. For each dataset, we evaluate our architecture on the original concept set used in the paper as well as four additional concept sets generated with GPT-5. All evaluations are run on every train–test split of each dataset. The table below shows the averaged accuracies with standard deviations across all five concept sets.
> > > >
> > > > |Backbone|Method|Breakfast|HMDB|UCF|SSv2|
> > > > |-|-|-|-|-|-|
> > > > |L/14|Global CBM|55.3±9.3|64.3±2.8|87.9±1.4|18.7±0.1|
> > > > |L/14|MoTIF|69.6±6.9|70.0±2.6|91.8±1.0|20.4±0.4|
> > > > |L/14|Global CBM|55.3±9.3|64.3±2.8|87.9±1.4|18.7±0.1|
> > > > |L/14|MoTIF-ST|71.5±6.6|72.0±1.9|92.6±0.8|24.6±0.7|
> > > > |PEL/14|Global CBM|72.7±9.1|70.9±2.2|93.1±1.0|25.8±0.3|
> > > > |PEL/14|MoTIF|84.0±5.7|76.2±2.2|95.3±0.6|29.6±0.3|
> > > > |PEL/14|Global CBM|72.7±9.1|70.9±2.2|93.1±1.0|25.8±0.3|
> > > > |PEL/14|MoTIF-ST|83.4±5.7|77.9±1.7|95.8±0.8|35.3±0.6|
> > > >
> > > > The main paper already includes an ablation on concept influence for Breakfast and HMDB with RN-50 and ViT-B/32. In our previous response, we also ablated concept composition on SSv2. Across all settings, concept choice does influence performance, consistent with prior CBM literature [1,2,3]. The results further indicate that our configuration was not tuned to a specific concept set, and that MoTIF behaves as expected under varying concept definitions. Concept extraction and concept-set design remain active research areas, with numerous studies aiming to improve the quality and usefulness of learned concepts. We believe that MoTIF offers a solid foundation for advancing concept-based modeling in the video domain and opens the door to further exploration in this direction.
> > > >
> > > > We hope that the proposed arguments and experiments address the reviewer's concerns. If so, we will include the remaining experiments as an additional subsection in the camera-ready version. Otherwise, we are open to further feedback.
> > > >
> > > > > Runtimes
> > > >
> > > > We reported runtimes only for our proposed approach because a fair comparison to existing methods is not feasible due to missing or unclear computational details in prior work. Our method combines (i) independently pre-trained foundation models as backbones and (ii) the training of MoTIF itself, whose cost is reported in Table 1&3.
> > > >
> > > > By contrast, existing video models require extensive pre-training phases. For example, VideoMAE V2 reports pre-training on UnlabeledHybrid for 241h on 64 A100 in parallel, and for the subsequent fine-tuning phase there are no computational cost provided (the repo suggests the use of 32 A100). In comparison, MoTIF requires only ~1h of training on a single A6000. Although a direct comparison is not possible, the difference in compute requirements is clearly substantial.
> > > >
> > > > > Performance–interpretability trade-off
> > > >
> > > > We agree that diagonal attention results in reduced accuracy compared to full attention, particularly on SSv2. However, we want to emphasize that this should not be interpreted solely as a performance decrease. Instead, it reflects a deliberate *performance–interpretability trade-off*. Diagonal attention ensures disentangled channels and prevents the model from forming arbitrary, visually opaque concepts that undermine interpretability (see Section B.2).
> > > >
> > > > ---
> > > >
> > > > We therefore kindly ask the reviewer to reconsider the performance assessment of our method in light of this trade-off, and to evaluate MoTIF within its intended scope - as an interpretable architecture that substantially improves upon the previously only available interpretable baseline [1].
> > > >
> > > > [3] Discover-then-Name, ECCV 2025
> > > >
> > > > [4] Chirality in Action, NeurIPS 2025

---

> > > > > ### Comment · Reviewer_y1P2 · 2025-11-26
> > > > >
> > > > > I thank the authors for the additional clarifications. The runtime argument sounds a bit odd to me since the question was about inference time, so I don't see how the other methods being slower to *train* prevents you from reporting their *inference* time.
> > > > >
> > > > > More importantly, however, I disagree with the key argument that interpretable models and effective models are two independent areas of research. In my opinion, there's very little practical value in developing highly interpretable models that perform dramatically worse than non-interpretable ones. It is hard for me to imagine a practical use case in which someone would choose to deploy such a model, sacrificing 50% of the performance for the sake of interpretability. Making actual sota models interpretable without sacrificing performance would be a lot more impactful. Like I said above, there's still purely academic value in this type of research, but the potential impact of it is limited, hence my score is only moderately negative and I would not mind this paper being accepted if other reviewers find it valuable.

---

> > > > > > ### Author Response · Authors · 2025-11-27
> > > > > >
> > > > > > We thank the reviewer for their continued feedback.
> > > > > >
> > > > > > ---
> > > > > >
> > > > > > > Inference vs. training runtimes
> > > > > >
> > > > > > We appreciate the clarification. Unfortunately, we cannot provide exact inference times for alternative methods due to missing or unclear details in their papers and public repositories (No frame left behind and VideoMAE V2 do not report inference times at all and TSM only reports inference times with parallel processing on SSv1).
> > > > > > MoTIF’s inference time per video is comparable to standard backbones (ViT or RN), since the temporal attention module adds minimal overhead relative to backbone extraction. The table below reports the absolute runtime overhead added by MoTIF and MoTIF-ST, along with the relative percentage overhead computed using the L/14 & PE-L/14 backbone runtimes. Across all datasets and backbones, MoTIF increases inference time by only 0.05–3.75%, confirming that temporal concept reasoning adds negligible computational cost relative to backbone extraction.
> > > > > >
> > > > > > |Backbone|Method|Breakfast|HMDB51|UCF101|SSv2|
> > > > > > |-|-|-|-|-|-|
> > > > > > |L/14|MoTIF|0.002s (0.33%)|0.002s (2.00%)|0.002s (1.43%)|0.002s (2.50%)|
> > > > > > |L/14|MoTIF-ST|0.004s (0.66%)|0.003s (3.00%)|0.003s (2.14%)|0.003s (3.75%)|
> > > > > > |PE-L/14|MoTIF|0.003s (0.05%)|0.002s (0.29%)|0.002s (0.23%)|0.002s (0.51%)|
> > > > > > |PE-L/14|MoTIF-ST|0.003s (0.05%)|0.003s (0.43%)|0.003s (0.35%)|0.003s (0.77%)|
> > > > > >
> > > > > > MoTIF-ST requires slightly more time due to the added temporal attention computations, but the overhead remains minimal relative to the backbone feature extraction.
> > > > > >
> > > > > > Furthermore, we report the total runtimes for MoTIF using the PE-L/14 backbone to provide an impression of the overall computational inference cost. The complete results for all backbones will be included in the appendix of the camera-ready version. Please note that all runtimes were measured without parallel batch processing (i.e., processing one video at a time) and on a single A6000 GPU.
> > > > > >
> > > > > > |Backbone|Breakfast|HMDB51|UCF101|SSv2|
> > > > > > |-|-|-|-|-|
> > > > > > |L/14|0.61s|0.10s|0.14s|0.08s|
> > > > > > |PE-L/14|5.54s|0.70s|0.86s|0.39s|
> > > > > >
> > > > > > > Performance vs. interpretability
> > > > > >
> > > > > > We thank the reviewer for the thoughtful comment. We agree that combining high performance with interpretability is desirable. While MoTIF does not yet match the raw performance of black-box SOTA models, it provides clear evidence that temporal structure can be learned and interpreted reliably—something appearance-based models cannot achieve. MoTIF serves as a proof-of-concept for interpretable temporal reasoning in videos, providing a foundation for future work aiming to integrate both strong performance and interpretability. We hope our results will motivate research bridging this gap.

---

### Official Review · Reviewer_qZ1Z · 2025-10-27

**Soundness:** 3
**Presentation:** 3
**Contribution:** 2
**Rating:** 4
**Confidence:** 4

**Summary:**

This paper introduces MoTIF (Moving Temporal Interpretable Framework), a new class of concept bottleneck models (CBMs) designed to provide interpretability for arbitrary-length video data. Unlike traditional CBMs that operate on static images or short clips, MoTIF extends the framework to handle long-term temporal dependencies, enabling interpretable reasoning across entire video sequences.

The model incorporates per-channel self-attention mechanisms to ensure that individual concept representations remain disentangled, preventing the mixing or interference of different semantic concepts across channels. Furthermore, MoTIF leverages a text-encoder-based grounding framework to tie each visual concept directly to natural language descriptions, linking model internals to human-understandable concepts. Additionally, MoTIF enables local-to-global interpretability: it allows fine-grained inspection of individual frame-level or concept-level activations, while also supporting high-level explanations of model behavior over full video trajectories.

**Strengths:**

- The paper tackles interpretability in the video domain, an area that remains significantly underexplored. Extending concept bottleneck models to temporal data represents an important and timely contribution.
- The proposed framework effectively supports probing at multiple levels of abstraction—from localized, concept-specific attention patterns to global, sequence-level explanations—offering a comprehensive view of model behavior.
- The paper is clearly written and well organized. The methodology, implementation details, and experimental setup are described with sufficient clarity and depth, making the work easy to follow and reproducible.

**Weaknesses:**

The core issue I have with the manuscript is the limited modeling of dynamics in the current version.

- Despite being framed as a video CBM, the method is largely appearance-centric. The concepts shown in the figures are predominantly static nouns or one-frame cues (e.g., bread/bagel in Breakfast; kayak/paddle in UCF101), and the explanation views mostly reweight per-window concept activations rather than modeling interactions that require multi-frame reasoning. The temporal module is per-channel self-attention that forbids cross-concept interactions, which may preserve attribution but weakens the capacity to capture genuinely dynamic, relational events (i.e., spatiotemporal object centric concepts, like a ball moving to the right). As a result, the paper does not convincingly demonstrate that the core challenge that makes video harder than images, *dynamics*, is addressed to a meaningful extent. If the main benefit is “interpretable CBMs that work on videos,” the paper should show that the model captures temporal structure that single frames cannot. As written, the strongest qualitative evidence highlights local/global concept scores and attention heatmaps, but the concepts themselves are mostly static and I am unconvinced that an image CBM would perform much worse or be less interpretable.

- Breakfast, HMDB51, UCF101 can be handled to a large degree with static cues; the model performs well there. The only dataset in the suite that truly requires temporal reasoning, SSv2, is where performance drops most sharply relative to strong non-interpretable baselines. Including non-interpretable SOTA numbers alongside Table 1 would make this gap clear; SSv2 state of the art is ~80% top-1 (e.g., MVD ≈77%), far above the reported results. This undercuts the claim that the method meaningfully tackles dynamics.

- Because concepts are LLM-generated without a video-centric constraint, there is no evidence the concept set contains temporal concepts that image CBMs would miss. It would be informative to (i) construct explicitly temporal concepts and show they are used, and (ii) demonstrate video-centric wins that image-centric CBMs cannot achieve. Looking at the concept list in Table 9, SSv2 is the only dataset that I see containing concepts which require more than one frame to understand - and is the only dataset with results far below non-interpretable baselines.

- The evaluation focuses on accuracy. There is no metric showing that the learned internal representations capture more dynamic information than a non-interpretable counterpart. Applying a dynamics-sensitivity analysis (e.g., using [a]) would directly test whether the model is learning temporal structure, not just aggregating static evidence over time.

Small things:

- Figure 6 caption should say UCF-101, not SSv2.
- Table 2 should compare against removing random concepts, and/or adding k concepts (insertion vs. deletion plots).
- Could the reason for better performance be a larger number of parameters used in the diagonal attention with many concepts compared to standard attention?

[a] Kowal et al. Quantifying and Learning Static vs. Dynamic Information in Deep Spatiotemporal Networks. (TPAMI 2024)

**Questions:**

- Compare against an image-CBM, like the one from Rao et al., 2024 and see if it does worse on temporal data/concepts. You could use the concepts in SSv2 that are known to be dynamic (e.g., pretending to touch smth, pushing smth left to right vs. right to left) and compare image CBMs to them, as well as the overall performance across the datasets.
- Alternatively, explicitly showing your method captures more dynamics than non-interpretable or image CBM baselines would be interesting (i.e., using [a] or frame shuffling experiments that show how a model is dependant on temporal ordering)
- Would it be possible to include explicitly temporal concepts (e.g., “starts to pour,” “moves backward”) in the concept generation process and assess whether MoTIF can meaningfully ground them? Again, showing MoTIF outperforms on these temporal concepts would be ideal.

---

> ### Author Response · Authors · 2025-11-18
> **Rebuttal (1/2)**
>
> We thank the reviewer for the constructive feedback and positive assessment of our work. We are grateful for the recognition that extending concept bottleneck models to temporal data represents an important and timely contribution to the underexplored area of video interpretability. The reviewers' appreciation for MoTIF's multi-level interpretability framework—supporting both fine-grained concept-level analysis and global sequence-level explanations—encourages us that our approach addresses a genuine need in the field.
>
> Below, we address the reviewers' specific concerns with additional experiments and analyses, with a particular focus on whether MoTIF truly captures temporal dependencies.
>
> First, a comment on SSv2 performance. Concurrent work that also applies CBMs to video data—but without temporal information [1] and accepted as a NeurIPS Spotlight—reports strong results on UCF101, which we outperform with our proposed architecture. However, they evaluate on comparatively “easier” datasets. We intentionally included SSv2 despite its lower absolute performance because this gap is important for advancing concept-based interpretability on challenging fine-grained datasets. As recent work shows [2], video embedders struggle heavily with distinguishing chirality-sensitive actions. Our hypothesis is that temporal-aware representations will substantially improve MoTIF and related approaches. The field is fortunately moving in this direction.
>
> ---
>
> > [...] see if it does worse on temporal data/concepts. You could use the concepts in SSv2 that are known to be dynamic (e.g., pretending to touch smth, pushing smth left to right vs. right to left)[...] Alternatively, explicitly showing your method captures more dynamics than non-interpretable or image CBM baselines would be interesting [...]
>
> Thanks for raising this point. We evaluate the proposed investigation in two complementary parts:
>
> **Frame Shuffling Analysis and Synthetic Temporal Pattern Experiment:** Before testing on real-world datasets, we first tested our hypothesis that MoTIF can learn temporal patterns using synthetic data. We designed a controlled experiment with 1,989 sequences (matching the number of instances in Breakfast) where class labels depend exclusively on frame ordering (ascending, descending, U-shaped, inverted U, periodic patterns).
> Results show **86.97% accuracy on original sequences** vs **21.06% on shuffled sequences** (near random chance for 5 classes). This **4.1× performance gap** demonstrates that MoTIF successfully learns temporal structure when it is required, validating our temporal attention mechanism's capability to capture order-dependent patterns.
> We also tested your suggested approach of shuffling the window embeddings on real-world datasets. As expected, these datasets do not strongly depend on strict temporal ordering—many actions are identifiable from short-term or static cues. Shuffling consistently causes a measurable but modest drop (averaged over five random seeds):
> - PE-L/14 Breakfast: 87.3% → 85.2–86.6%
> - PE-L/14 HMDB: 79.9% → 77.1–78.5%
> - PE-L/14 UCF101: 94.7% → 94.5–94.6%
> - PE-L/14 SSv2: 30.0% → 26.9–27.4%
>
> These results jointly show that MoTIF can learn temporal structure when required (synthetic experiment), but most real-world benchmarks, except SSv2, do not make strict temporal dependencies necessary for achieving high accuracy.
>
> **Comparison with Image-CBM Baseline:** We compared MoTIF against an Image-CBM baseline that aggregates all frame embeddings and produces a single, global concept activation per video. Previously, we referred to this as a “linear probe,” but to avoid confusion we now call this method the Global CBM (see updated paper). On our synthetic temporal pattern dataset, the Global CBM achieves only **35.5% accuracy**, significantly lower than MoTIF's 86.97%, demonstrating that temporal modeling is essential when order matters. We also evaluated both methods on temporal concepts from SSv2. MoTIF shows superior performance on these dynamic concepts compared to the Global CBM baseline, which struggles to distinguish time-asymmetric actions (see Table 1 in the updated paper).

---

> > ### Author Response · Authors · 2025-11-18
> > **Rebuttal (2/2)**
> >
> > > Would it be possible to include explicitly temporal concepts (e.g., "starts to pour," "moves backward") in the concept generation process and assess whether MoTIF can meaningfully ground them? Again, showing MoTIF outperforms on these temporal concepts would be ideal.
> >
> > To address whether MoTIF can meaningfully ground explicitly temporal concepts, we created five concept sets for SSv2, which most relies on such temporal concepts: (1) nouns only, (2) verbs only, (3) nouns combined with verbs, (4) the original concept set used in our paper, and (5) all concepts together (see updated appendix for full concept lists and Table 1). We tested these with standard MoTIF-PE/L14 and the updated TimesFormer architecture (addressed by reviewer [y1P2]), which consistently outperforms Global CBM across all concept sets.
> >
> > **Table 1: MoTIF concept ablation on SSv2**
> >
> > | Concept Set | MoTIF |MoTIF-ST | Global CBM |
> > |-|-|-|-|
> > | (1) Nouns only | 18.0% |20.3% | 14.8% |
> > | (2) Verbs only | 24.4% |28.6% | 21.2% |
> > | (3) Nouns + Verbs | 22.1% |24.5% | 18.2% |
> > | (4) Original set | 30.0% |35.1% | 25.5% |
> > | (5) All concepts | 29.8% |36.0% | 26.7% |
> >
> > The results demonstrate several key findings: (1) MoTIF consistently outperforms Global CBM across all concept sets, confirming that temporal modeling provides substantial benefits. (2) Verb-only concepts (set 2) show higher performance than noun-only (set 1), indicating that MoTIF excels at capturing action dynamics that inherently require temporal understanding. (3) The original concept set (set 4) achieves the highest performance (37.8%), suggesting that a balanced mix of concepts is optimal. Across all concept sets, MoTIF improves over Global CBM, showing that MoTIF’s temporal modeling reliably strengthens concept grounding regardless of the chosen vocabulary.
> >
> > > Table 2 should compare against removing random concepts, and/or adding k concepts (insertion vs. deletion plots).
> >
> > Thank you for raising this point. We performed two additional types of concept interventions: (i) random concept removal (removing $k$ randomly selected concepts), and (ii) random insertion (adding Gaussian noise $\epsilon \sim \mathcal{N}(0, 0.5^2)$ to $k$ randomly selected concept channels). Table 2 in shows the normalized accuracy (relative to baseline) after each intervention for Breakfast and HMDB51.
> >
> > **Table 2: Expanded concept intervention study**
> >
> > | Dataset | $k$ | Top-$k$ Removal | Random Removal | Random Insertion |
> > |-|--|-|---|-|
> > | **Breakfast** | 0 | 1.000 | 1.000 | 1.000 |
> > | | 1 | 0.496 | 0.989 | 0.979 |
> > | | 2 | 0.229 | 0.972 | 0.947 |
> > | | 3 | 0.085 | 0.951 | 0.951 |
> > | | 4 | 0.028 | 0.923 | 0.908 |
> > | **HMDB51** | 0 | 1.000 | 1.000 | 1.000 |
> > | | 1 | 0.603 | 0.974 | 0.978 |
> > | | 2 | 0.374 | 0.963 | 0.961 |
> > | | 3 | 0.238 | 0.949 | 0.945 |
> > | | 4 | 0.142 | 0.935 | 0.921 |
> >
> > The results demonstrate that MoTIF's predictions depend on a small, semantically meaningful subset of concepts. When removing the top-$k$ most influential concepts, accuracy drops dramatically (e.g., Breakfast: 49.6% at $k=1$, 2.8% at $k=4$). In contrast, random concept removal causes minimal degradation (Breakfast: 98.9% at $k=1$, 92.3% at $k=4$), and random noise insertion shows similar resilience (Breakfast: 97.9% at $k=1$, 90.8% at $k=4$). This sharp contrast between top-$k$ removal and random interventions confirms that MoTIF identifies and relies on specific, meaningful concepts rather than distributed, redundant activations. In addition, the random noise insertion further confirms that MoTIF remains robust to noise.
> >
> > > Could the reason for better performance be a larger number of parameters used in the diagonal attention with many concepts compared to standard attention?
> >
> > As shown in Figure 4, MoTIF with diagonal attention actually performs slightly worse (or at best comparable) compared to standard cross-channel attention across our datasets. The diagonal attention mechanism is designed not for performance gains, but rather for interpretability—it ensures that individual concept representations remain disentangled by preventing cross-channel mixing, which is essential for meaningful concept-level explanations. We would appreciate clarification if we misunderstood the question.
> >
> > **Minor remarks:**
> > - We updated Table 6.
> > - All code and results are in the updated PDF and supplementary material (code).
> >
> > [1] Disentangled Concepts Speak Louder Than Words: Explainable Video Action Recognition, NeurIPS 2025
> >
> > [2] Chirality in Action: Time-Aware Video Representation Learning by Latent Straightening, NeurIPS 2025
> >
> > ---
> >
> > We are truly thankful for this valuable feedback, especially for highlighting the importance of showing in more detail that MoTIF really learns temporal dependencies. We hope that we have answered your questions and were able to convince you that MoTIF performs better than normal CBMs. We would be happy to further address any remaining questions during the remainder of the rebuttal period.
> >
> > Best regards,
> > MoTIF authors

---

> > > ### Comment · Reviewer_qZ1Z · 2025-11-26
> > >
> > > Thank you for the significant response and additional experimental results!
> > >
> > > As for [1], they indeed use HMDB but output a pose-based interpretability visualization (SSv2 doesn't have the full pose in every video anyway) specifically and explicitly disentangle motion vs. object vs. scene information in their architecture. This is a significant difference from image-based CBMs and in my mind a major contribution which might be why they got a spotlight at NeurIPS. While I took a quick look at [1] and it seems like a solid paper, the fact that it was accepted (and whatever awards it happens to receive) does not guarantee that [1]'s, and therefore this papers, experimental settings was sound. Overall, this argument is unconvincing and I believe novel methods for interpreting video must deal with the temporal information differently/better/in a novel way compared to image-based baselines, and appearance-based HMDB/UCF101 etc are insufficient to evaluate video interpretability methods.
> > >
> > > To this end, I appreciate the further experimental results. I think the top-k removal experiments are convincing and should be included in the final paper (even if in supplemental material), as it shows the concepts are more important to the output than randomly selected ones. With regards to temporal experiments, I am unsurprised that shuffling doesn't drop performance much, but for SSv2 it does quite a bit for models that achieve strong performance (e.g., see the shuffling results from [a]). Additionally, the "Table 1: MoTIF concept ablation on SSv2" results are great - these should definitely be in the main paper in my opinion as clear evidence that your proposed method captures dynamics in the CBM layer better than image-based counterparts. However, I would emphasize that the authors include a limitation in the "Several open challenges remain" section highlighting that the current performance of ALL video CBMs is extremely poor compared to non-interpretable counterparts for datasets like SSv2 (i.e., we are nowhere close to solving the problem, 36 - 80% is a massive gap). So the current approach is the best, but not final, for dynamic concepts.
> > >
> > > Finally, I would like to know more details about the synthetic experiment. Is it possible to describe in more detail what these videos are and how the model is trained? What is the task exactly? I think this could also be worth including in the supplemental materials too in my opinion, especially if you could compare against image-based CBMs (which should get random performance if my understanding is correct).
> > >
> > > If you can simply provide some more details about the synthetic experiment, I think I would be ready to raise my score. While I still push back on some of the claims made, e.g., that MOTIF is similar to [1] in terms of dynamic explanations, and that MOTIF can do a sufficient job at explaining temporal concepts (it does the *best* job on SSv2, but far below where we want to be to truly understand video models), I think the updated paper and evidence puts the method on much more solid ground.

---

> > > > ### Author Response · Authors · 2025-11-27
> > > >
> > > > We thank the reviewer for the positive consideration of our rebuttal and for the additional feedback.
> > > >
> > > > ---
> > > >
> > > > > [...] appearance-based HMDB/UCF101 etc are insufficient to evaluate video interpretability methods.
> > > >
> > > > We include HMDB/UCF101 datasets for comparability with prior work [1]. However, we agree that appearance-centric datasets are limited for evaluating temporal interpretability. This is precisely why we introduced experiments that isolate temporal structure (SSv2, synthetic data, shuffled-sequence ablations), going beyond what image-based CBMs or appearance-heavy datasets can reveal. We hope this encourages future research to adopt datasets and tasks that require genuine temporal understanding.
> > > >
> > > > > Paper modification
> > > >
> > > > We are pleased that the top-k removal experiments were convincing and have already integrated them into the updated version. Regarding the shuffling experiment, we agree that temporal dependencies in datasets other than SSv2 are weaker and therefore less conclusive. Nevertheless, we will include the results in the main paper to demonstrate that MoTIF does exploit temporal structure when the dataset requires it (so also the synthetic dataset). For the limitations section, we will add the suggested discussion. Our goal is not to present MoTIF as a complete solution, but as a step toward closing the gap between model performances that still exists for complex datasets such as SSv2.
> > > >
> > > > > Synthetic data
> > > >
> > > > We are happy to hear that the synthetic data experiment arouses interest. For full transparency, we already uploaded the corresponding code in our previous response (supplementary material: Synthetic_temporal_pattern.ipynb). Following the reviewer’s suggestion, we will integrate the key results directly into the main paper and place the full synthetic data description in the appendix.
> > > >
> > > > The synthetic experiment is designed to isolate temporal reasoning by removing all appearance cues. Each “video” is a sequence of length-N (matching the Breakfast distribution), where each frame is a 1024-dimensional vector. We define five temporal classes: (1) ascending, (2) descending, (3) U-shape, (4) inverted-U, and (5) periodic oscillation. Class identity is determined only by the temporal trajectory of 30% of the dimensions; the remaining dimensions are filled with noise. Thus, the task can only be solved by learning the temporal order—static appearance information is uninformative by design.
> > > >
> > > > To examine whether the model can learn interpretable temporal structure, we provide 37 artificial temporal pattern concepts. Each concept represents a prototypical analytic temporal signal (ascending, descending, S-curve, exponential growth/decay, chirp, sawtooth, step-up/down, periodic-fast, oscillating-decay, double-peak, plateau, etc.). All embeddings encode temporal shape only, without semantics.
> > > >
> > > > The training setup matches our real pipelines, allowing a direct test of MoTIF’s temporal inductive biases. MoTIF achieves 86.97% accuracy on the original (ordered) sequences but only 21.06% when frames are randomly permuted (chance ≈ 20%). This 4.1× drop shows that MoTIF relies on temporal ordering rather than static frame statistics. An image-level CBM baseline (“Global CBM”), which collapses all frames into a single vector, reaches only 35.5%, confirming that methods without temporal modeling cannot solve this task.
> > > >
> > > > As an additional control, we trained MoTIF on permanently shuffled sequences. As expected, performance collapsed to the level of the image-level CBM baseline (“Global CBM”), reaching only \~40%. We attribute this to the fact that when temporal order is removed (shuffled inputs) or cannot be exploited by design (Global CBM), the model is forced to rely on residual non-temporal cues—such as residual global statistics (e.g., small differences in overall magnitude or variance) that survive shuffling—that provide weak but above-chance signal. While such non-temporal cues can nudge performance above the 20% chance baseline, they are inherently insufficient for achieving meaningful temporal classification. In contrast, when temporal structure is preserved and explicitly modeled, MoTIF leverages its temporal inductive bias to achieve substantially higher accuracy (\~87%), highlighting the importance of temporal reasoning for this task. Thereby, the reliance on temporal information becomes evident, since shuffling only at evaluation time causes accuracy to collapse to chance level.
> > > >
> > > > ---
> > > >
> > > > We thank the reviewer once again for the constructive discussion and for helping us improve the paper. We would greatly appreciate it if the reviewer would consider raising their score as indicated. If there are any remaining concerns or questions, we are of course happy to address them.

---

### Official Review · Reviewer_8q69 · 2025-10-31

**Soundness:** 2
**Presentation:** 3
**Contribution:** 3
**Rating:** 6
**Confidence:** 2

**Summary:**

This paper introduces MoTIF (Moving Temporal Interpretable Framework), a novel architecture for interpretable video classification. The work extends the principles of Concept Bottleneck Models (CBMs), which have been successful for images, to the video domain. The core challenge in this extension is modeling temporal dependencies while maintaining the interpretability of individual concepts. MoTIF addresses this with a transformer-inspired architecture that employs a "per-channel temporal self-attention" mechanism. This mechanism intentionally prevents the mixing of concept channels, allowing the model to reason about the temporal evolution of each concept independently.

**Strengths:**

- The primary contribution is the adaptation of the concept bottleneck paradigm to sequential data like video. The proposed "diagonal attention" mechanism, which uses depthwise 1x1 convolutions to create per-concept QKV projections, is a clever and effective way to model temporal dynamics without sacrificing concept-level interpretability.
- The framework is explicitly designed to produce three distinct and useful types of explanations: global, local, and temporal. This provides a much richer and more comprehensive view of the model's decision-making process than a single vector of concept importance.
- The paper presents a comprehensive set of experiments across four diverse datasets and a range of modern vision-language backbones.

**Weaknesses:**

- A core design choice of MoTIF is to enforce strict concept isolation via diagonal attention to maintain interpretability. However, many complex actions are inherently compositional, defined by the interactions between different concepts over time, which MoTIF cannot capture.

**Questions:**

This paper presents a novel and well-executed contribution.

- In Figure 5, what is the white circle?

---

> ### Author Response · Authors · 2025-11-18
> **Rebuttal (Part 1 / 1)**
>
> We sincerely thank the reviewer for the very positive and encouraging feedback, especially for recognizing the novel diagonal attention mechanism and the three complementary explanation views evaluated across multiple datasets and backbones.
>
> Below, we address the reviewer's questions and comments in detail.
>
> ---
>
> > In Figure 5, what is the white circle?
>
> Figure 5 depicts an Inter-Quartile-Range (IQR) plot. Each dot represents a single MoTIF instance with a different random seed; the white circle marks an outlier. We have clarified this in the revised manuscript to improve the figure's interpretability.
>
> > A core design choice of MoTIF is to enforce strict concept isolation via diagonal attention to maintain interpretability. However, many complex actions are inherently compositional, defined by the interactions between different concepts over time, which MoTIF cannot capture.
>
> We fully agree with this observation. A similar concern was raised by Reviewer [y1P2]. While the original diagonal attention enforces concept isolation, it limits the model's ability to capture interactions between concepts across time. In response, we extended MoTIF with a space–time attention mechanism inspired by transformer-based video models [1] (see Table 1). Space–time attention allows the model to selectively attend to multiple concept channels and their temporal evolution, enabling it to model compositional actions without losing interpretability. This modification improves performance consistently across all datasets. As a result, MoTIF now approaches closer to non-interpretable baselines while maintaining its interpretability.
>
> **Table 1: MoTIF with Space-Time Transformers**
>
> |Backbone|Breakfast|HMDB51|UCF101|SSv2|
> |-|-|-|-|-|
> |MoTIF-ST (L/14)|73.5 +/- 12.2| 74.8 +/- 1.0|93.8 +/- 0.9|23.9|
> |MoTIF-ST (PE-L/14)|83.6 +/- 7.0|79.6 +/- 0.7|96.3 +/- 0.6|35.1|
>
> All experimental details and additional results are included in the revised manuscript.
>
> [1] Is Space-Time Attention All You Need for Video Understanding?, ICML 2021
>
> ---
>
> We hope this clarifies our design decisions and that the added experiments with the space–time transformer extension address your concerns. We would be happy to provide further details during the remainder of the rebuttal period if you have additional questions.
>
> Best regards,
> MoTIF Authors

---

> > ### Comment · Reviewer_8q69 · 2025-11-28
> >
> > Thank you for your response. I will keep my original recommendation.

---

### Official Review · Reviewer_nyqo · 2025-11-04

**Soundness:** 4
**Presentation:** 3
**Contribution:** 4
**Rating:** 8
**Confidence:** 3

**Summary:**

The paper introduces MoTIF (Moving Temporal Interpretable Framework), an architectural design that successfully adapts the Concept Bottleneck Model (CBM) framework for video classification, handling sequences of arbitrary length. CBMs previously struggled with videos due to inherent temporal dependencies and the variable length of video data. MoTIF addresses this by incorporating transformer-inspired blocks while strictly preserving concept separation.  The primary innovation is the Per-channel temporal self-attention (diagonal attention) mechanism, which isolates temporal reasoning for each concept channel using depthwise 1x1 convolutions. MoTIF explicitly models reoccurring semantic entities (motifs) and provides three complementary explanation views: global concept importance, local concept relevance within specific temporal windows, and temporal dependency maps visualized through attention heads. Experiments on four benchmarks (Breakfast Actions, HMDB51, UCF101, and Something-Something V2) demonstrate that MoTIF achieves competitive performance while offering fine-grained and faithful explanations of concept contributions in temporal contexts.

**Strengths:**

MoTIF successfully extends the CBM principle from static images to sequential data. This innovation bridges the gap between interpretability and high-capacity temporal architectures

The core mechanism, per-channel temporal self-attention (diagonal attention), maintains concept independence by avoiding cross-channel mixing in the QKV projections.  Ablation studies confirm that MoTIF preserves generalization while enforcing interpretability.

MoTIF offers three crucial and complementary explanation views: (i) global concept relevance (via LSE pooling), (ii) localized temporal explanations (using windowed attributions, identifying concepts active in decisive windows), and (iii) attention-based temporal maps that visualize how each concept channel focuses across time, exposing anchor moments or consistent presence.

MoTIF achieves competitive predictive performance, surpassing zero-shot and linear-probe baselines across all four datasets. Performance improves consistently with higher capacity CLIP-based backbones

**Weaknesses:**

The architecture deliberately trades computational efficiency for strict concept isolation; thus, when the number of concepts (C) is large, full MHA may be more efficient. Disabling diagonal attention significantly reduced GPU memory usage and epoch time for longer videos, such as those in the Breakfast Actions dataset

The architectural design suppresses cross-concept interactions to maintain interpretability. Consequently, stacking more blocks offers limited or no richer hierarchical abstractions, which deliberately restricts the model’s expressivity compared to conventional deep transformers

MoTIF maintains competitive accuracy given its interpretability constraint. However, on challenging datasets like SSv2, a performance gap persists when compared to strong non-interpretable baselines such as VideoMAE V2.

 Open challenges remain regarding key non-trivial design choices, such as selecting the optimal window size, which depends on action duration, speed, and alignment

**Questions:**

How do the authors ensure the practicality and deployment efficiency of MoTIF when working with very long video sequences and a rich concept vocabulary?

Given that performance can be sensitive to the concept set, what are the recommendations for extracting richer and more action-relevant concepts directly from video data, especially for abstract classes like those found in SSv2, rather than relying solely on external language model guidance?.

---

> ### Author Response · Authors · 2025-11-18
> **Rebuttal (Part 1 / 2)**
>
> We would like to thank the reviewer for the very positive and encouraging feedback — especially for recognizing that MoTIF successfully bridges the gap between Concept Bottleneck Models and temporal models, as well as for acknowledging the diagonal attention mechanism and three complementary explanation views, which demonstrate strong performance across several established baselines.
>
> Below, we address the questions and points raised by the reviewer:
>
> ---
>
> > How to ensure practicality and deployment efficiency of MoTIF when working with very long video sequences and a rich concept vocabulary?
>
> MoTIF’s computational complexity grows with both the number of concepts and the length of input videos. In our current experiments, we successfully train models on videos up to 11 minutes long (Breakfast dataset). For longer videos or larger concept vocabularies, efficiency can be improved by applying a sliding window approach: the video is split into shorter segments, MoTIF is applied to each segment independently, and the resulting segment-level embeddings are aggregated—e.g., via a linear classifier—to produce the final prediction. This strategy could allow MoTIF to scale to very long sequences while maintaining interpretability and practical deployment efficiency.
>
> > [...] what are the recommendations for extracting richer and more action-relevant concepts directly from video data [...].
>
> MoTIF was intentionally designed to be compatible with many existing and emerging concept extraction techniques used in CBMs. This includes foundation model–based approaches ([1], [2]) and automated discovery frameworks ([3]).
>
> A key factor for successful concept extraction is ensuring adequate coverage of action-relevant dynamics. For instance, [1] introduces motion activity concepts in a non-transformer-based CBM, concurrent to our work (accepted as Spotlight at NeurIPS 2025). Although MoTIF does not primarily target concept extraction, it already achieves strong performance on shared benchmark datasets. Integrating MoTIF with more advanced concept discovery methods, such as those proposed in [1], could further improve video understanding in future work.
>
> > The architectural design suppresses cross-concept interactions to maintain interpretability.
>
> We agree this property, albeit restrictive in cross-concept interactions, enables temporal interpretability while ensuring that each concept remains semantically isolated. As noted in the paper, MoTIF also aims to inspire future research toward models that can balance interpretability and richer cross-concept abstraction (see Section 5).

---

> ### Author Response · Authors · 2025-11-18
> **Rebuttal (Part 2 / 2)**
>
> > MoTIF maintains competitive accuracy [...]. However, on challenging datasets like SSv2, a performance gap persists [...]
>
> Following this comment (and that of Reviewer [y1P2]), we expanded MoTIF with space-time inspired attention mechanisms [5], improving performance consistently across all datasets (see Table 1). Additionally, the concept-type ablation study (raised by [qZ1Z]) revealed that refining concept categories further boosts SSv2 results (see Table 2).
>
> MoTIF is now closer to non-interpretable baselines while preserving transparency. The updated scores are listed below.
>
> **Table 1: MoTIF with space-time tranformers**
>
> |Backbone|Breakfast|HMDB51|UCF101|SSv2|
> |-|-|-|-|-|
> |MoTIF-ST (L/14)|73.5 +/- 12.2| 74.8 +/- 1.0|93.8 +/- 0.9|23.9|
> |MoTIF-ST (PE-L/14)|83.6 +/- 7.0|79.6 +/- 0.7|96.3 +/- 0.6|35.1|
>
> **Table 2: MoTIF concept ablation on SSv2**
>
> |Concept Set|MoTIF|MoTIF-ST|Global CBM|
> |-|-|-|-|
> | (1) Nouns only | 18.0% |20.3% |14.8% |
> | (2) Verbs only | 24.4% |28.6% |21.2% |
> | (3) Nouns + Verbs | 22.1% |24.5% | 18.2%|
> | (4) Original set | 30.0% |35.1% |25.5%|
> | (5) All concepts | 29.8% |36.0% |26.7%|
>
> All details and new results are provided in the Appendix (new Section) of the revised version.
>
> Concurrent work that also applies CBMs to video data—but without temporal information [1] and accepted as a NeurIPS Spotlight—reports strong results on UCF101, which we outperform. However, they evaluate on comparatively “easier” datasets. We intentionally included SSv2 despite its lower absolute performance because this gap is important for advancing concept-based interpretability on challenging fine-grained datasets. As recent work shows [4], video embedders struggle heavily with distinguishing chirality-sensitive actions. Our hypothesis is that temporal-aware representations will substantially improve MoTIF and related approaches. The field is fortunately moving in this direction.
>
> [1] Disentangled Concepts Speak Louder Than Words: Explainable Video Action Recognition, NeurIPS 2025
>
> [2] DCBM: Data-Efficient Visual Concept Bottleneck Models, ICML 2025
>
> [3] Discover-then-Name: Task-Agnostic Concept Bottlenecks via Automated Concept Discovery, ECCV 2025
>
> [4] Chirality in Action: Time-Aware Video Representation Learning by Latent Straightening, NeurIPS 2025
>
> [5] Is Space-Time Attention All You Need for Video Understanding?, ICML 2021
>
> ---
>
> We hope this clarifies your questions and we would be happy to provide further details during the remainder of the rebuttal period if you have additional questions or comments.
>
> Best regards,
> MoTIF Authors

---

### Author Response · Authors · 2025-11-18
**Rebuttal Summary**

We thank all reviewers for their positive and constructive feedback. We appreciate the recognition that the paper is clearly written, well organized, and easy to follow, and that the methodological design is sound [y1P2, qZ1Z, nyqo]. Reviewers emphasized that extending concept bottleneck models to the video domain addresses an important and underexplored problem, and noted that MoTIF’s temporal modeling capabilities [nyqo, 8q69, qZ1Z], per-channel attention mechanisms, and multi-level explanations form a coherent and technically well-motivated framework [nyqo, 8q69, y1P2]. We are pleased that reviewers found the contribution timely and that the clarity of the implementation details and experimental setup supports reproducibility.

All newly added experiments are highlighted in blue in the revised PDF, and the adapted codebase has been uploaded. Below, we summarize the three most important improvements addressed in this rebuttal:
- Performance on SSv2. We directly compare MoTIF against a concurrent approach [1] and additionally report results from recent analyses [2] showing why video embedders struggle specifically on SSv2. These findings help explain why MoTIF, which builds on these embedders, is similarly affected.
- Extended temporal concept experiments. We expand our analysis of temporal concepts through (i) a synthetic dataset where classification is only possible if temporal order is learned, (ii) multiple concept-set variants, and (iii) an ablation that shuffles window embeddings. All results consistently confirm that MoTIF learns temporally grounded concepts.
- MoTIF with Space-Time Transformers. We demonstrate that MoTIF integrates cleanly with other transformer architectures, including the SpaceTime Transformer [3], achieving a +5.1% improvement on SSv2.

[1] Disentangled Concepts Speak Louder Than Words: Explainable Video Action Recognition, NeurIPS 2025

[2] Chirality in Action: Time-Aware Video Representation Learning by Latent Straightening, NeurIPS 2025

[3] Is Space-Time Attention All You Need for Video Understanding?, ICML 2021

---

Best regards,

MoTIF Authors

---

### Author Response · Authors · 2025-12-01
**Rebuttal Summary for AC**

We thank the reviewers for their constructive feedback during the rebuttal phase and appreciate the opportunity to clarify and strengthen our work. We chose ICLR specifically because of its emphasis on meaningful author–reviewer interaction, and we have always had positive experiences with the discussion phase. Unfortunately, due to the security incident, this year’s rebuttal period ended abruptly. Below we summarize the key outcomes of the (very positive) discussion. We also thank both the outgoing and incoming AC for handling this unusual situation.

Across the board, our paper was received positively. Reviewers [nyqo], [8q69], and [qZ1Z] expressed support from the beginning. Reviewer [qZ1Z] initially questioned whether MoTIF truly learns temporal patterns; in response, we conducted a synthetic experiment (now included in the supplementary material). The reviewer indicated that this experiment addressed the concern and expressed to raise the score, contingent only on a more detailed description of the setup—an update that the premature shutdown unfortunately prevented. The reviewer also agreed that our concept-intervention experiments clearly demonstrate MoTIF’s ability to identify which concepts drive the prediction.

Reviewer [y1P2] likewise provided positive feedback and requested additional experiments that we were able to complete during the rebuttal period. These results showed that MoTIF integrates cleanly into space–time transformer architectures; that performance remains within the expected range even when sampling different concept sets; that diagonal attention ablations on SSv2 behave as anticipated; and that MoTIF adds only 0.05–3.75% inference overhead relative to backbone feature extraction—meaning temporal concept reasoning is computationally negligible.

The remaining concern was that MoTIF does not yet match the strongest non-interpretable baselines on very challenging datasets such as SSv2. However, we directly compare MoTIF against a concurrent approach [1] and additionally report results from recent analyses [2] showing why video embedders struggle specifically on SSv2. These findings help explain why MoTIF, which builds on these embedders, is similarly affected. Our goal, however, is to establish a new architectural direction that enables faithful temporal concept reasoning while keeping performance competitive, thereby opening a path for future work to close this gap.

Overall, we believe the rebuttal phase allowed us to thoroughly and convincingly address all raised questions. As said, we will integrate the whole feedback in the camera-ready version.

Best,

The MoTIF Authors

---

[1] Disentangled Concepts Speak Louder Than Words: Explainable Video Action Recognition, NeurIPS 2025

[2] Chirality in Action: Time-Aware Video Representation Learning by Latent Straightening, NeurIPS 2025

---

### Meta-Review · Area_Chair_V7bw · 2025-12-28

**Summary:**

The paper introduces Moving Temporal Interpretable Framework, a transformer-inspired extension of Concept Bottleneck Models to video classification that captures temporally recurring semantic concepts and their dependencies over time. The framework provides global, local, and temporal concept-level interpretability while maintaining competitive performance on video understanding tasks.

This submission is recommended for rejection based on two central issues raised consistently by the reviewers.

First, Reviewer qZ1Z highlights that the method does not meaningfully capture video dynamics: the learned concepts are largely static, cross-concept temporal interactions are explicitly forbidden, and performance drops sharply on SSv2—the dataset most dependent on temporal reasoning. After the rebuttal, the authors’ results themselves indicate that the approach relies primarily on spatial, frame-level cues rather than modeling genuine temporal interactions, making the contribution incremental and insufficient to address the core challenges of video understanding beyond image-based CBMs.

Second, Reviewer y1P2 and Reviewer nyqo raise closely related concerns regarding the interpretability–performance trade-off. While accuracy remains competitive on simpler datasets, a substantial performance gap persists on challenging benchmarks like SSv2, calling into question the practical value of sacrificing performance for interpretability. Without demonstrating that the interpretability gains meaningfully enable better understanding, debugging, or deployment in real-world settings, it remains unclear why such a trade-off is justified, limiting the impact and applicability of the proposed approach. AC agrees with both reviewers on this point and does not see a clear or feasible path to quickly addressing these fundamental limitations within the scope of the current submission.

**Reviewer Concerns:**

Reviewer nyqo:
The reviewer notes that MoTIF deliberately sacrifices efficiency and expressivity to preserve strict concept isolation, which limits scalability and hierarchical abstraction compared to standard transformers. While accuracy remains competitive, a performance gap persists on challenging datasets like SSv2, and key design choices such as window size and concept vocabulary remain unresolved. They also question practicality for long videos and how to extract richer, action-relevant concepts directly from video data.

Reviewer 8q69:
The reviewer is concerned that strict concept isolation prevents MoTIF from modeling compositional actions that require interactions between concepts over time. Although they find the contribution novel and well executed, this design choice fundamentally limits expressivity for complex actions.

Reviewer qZ1Z:
The main concern is that the method does not meaningfully capture video dynamics, as concepts are largely static and cross-concept temporal interactions are forbidden. Performance drops sharply on SSv2—the dataset most reliant on temporal reasoning—calling into question whether the approach addresses the core challenges of video understanding. The reviewer requests stronger evidence that MoTIF captures temporal structure beyond what image-based CBMs can provide.

Reviewer y1P2:
The reviewer argues that reliance on frame-level representations severely limits spatiotemporal modeling, leading to very poor performance on SSv2 compared to state-of-the-art space-time transformers. They question the method’s real-world impact without results on modern spatiotemporal backbones and non-toy datasets. Additional concerns include missing runtime analysis, variance reporting, and insufficient ablations of diagonal attention on strong architectures.

The reviewers’ concerns are partially addressed through the extensive additional experiments provided by the authors during the rebuttal.

**Reviewer Scores:**

The initial scores before the rebuttal were 8, 6, 4, and 4. After the rebuttal, Reviewer 8q69 maintains their score, and Reviewer qZ1Z indicates an intention to increase their score; however, the core concerns outlined above remain unresolved. Reviewer y1P2 maintains a score of 4 due to these concerns, which the AC finds valid, and similar issues are also raised by Reviewer nyqo despite their higher score of 8. Taken together, and given the persistence of these fundamental concerns despite some positive scores, the AC determines that the paper should be rejected.

---

### Decision · Program_Chairs · 2026-01-26

Reject